# Promptable 3-D Object Localization with Latent Diffusion Models

**Cheng-Yao Hong**[1]    **Li-Heng Wang**[1,2]    **Tyng-Luh Liu**[1]

[1]Institute of Information Science, Academia Sinica [2]University of Southern California

`{sensible, liheng, liutyng}@iis.sinica.edu.tw`

## Abstract

Accurate identification and localization of objects in 3-D scenes are essential for advancing comprehensive 3-D scene understanding. Although diffusion models have demonstrated impressive capabilities across a broad spectrum of computer vision tasks, their potential in both 2-D and 3-D object detection remains underexplored. Existing approaches typically formulate detection as a "noise-to-box" process, but they rely heavily on direct coordinate regression, which limits adaptability for more advanced tasks such as grounding-based object detection. To overcome these challenges, we propose a promptable 3-D object recognition framework, which introduces a diffusion-based paradigm for flexible and conditionally guided 3-D object detection. Our approach encodes bounding boxes into latent representations and employs latent diffusion models to realize a "**promptable noise-to-box**" transformation. This formulation enables the refinement of standard 3-D object detection using textual prompts, such as class labels. Moreover, it naturally extends to grounding object detection through conditioning on natural language descriptions, and generalizes effectively to few-shot learning by incorporating annotated exemplars as visual prompts. We conduct thorough evaluations on three key 3-D object recognition tasks: general 3-D object detection, few-shot detection, and grounding-based detection. Experimental results demonstrate that our framework achieves competitive performance relative to state-of-the-art methods, validating its effectiveness, versatility, and broad applicability in 3-D computer vision.

## 1 Introduction

Precise identification and accurate localization of objects constitute foundational tasks critical for advancing the interpretation and analysis of visual data within computer vision. While numerous methodologies [54, 55] have achieved remarkable outcomes in traditional 2-D settings, exemplified by precise and efficient real-time detection in image domains, these approaches have increasingly encountered performance plateaus. In contrast, object detection within 3-D environments introduces substantial complexity, posing additional challenges that render direct adaptation of successful 2-D methodologies inadequate for handling the intricate dynamics of 3-D data.

Recent advancements have leveraged diffusion-based models, showcasing considerable promise in refining and enhancing solutions to conventional computer vision tasks. In particular, object detection has benefited from these developments, as evidenced by the pioneering work of DiffusionDet [7], which introduced the concept of diffusion processes as "noise-to-box" transformations within detection frameworks. Subsequent extensions have successfully applied analogous diffusion concepts to 3-D object detection, validating the versatility of diffusion methods across both 2-D and 3-D modalities. Nevertheless, prevailing diffusion-based detection methodologies predominantly employ the classical diffusion process, directly predicting the target outputs rather than noise distributions, and operate entirely within the original feature space. Consequently, such methods exhibit limited

39th Conference on Neural Information Processing Systems (NeurIPS 2025).

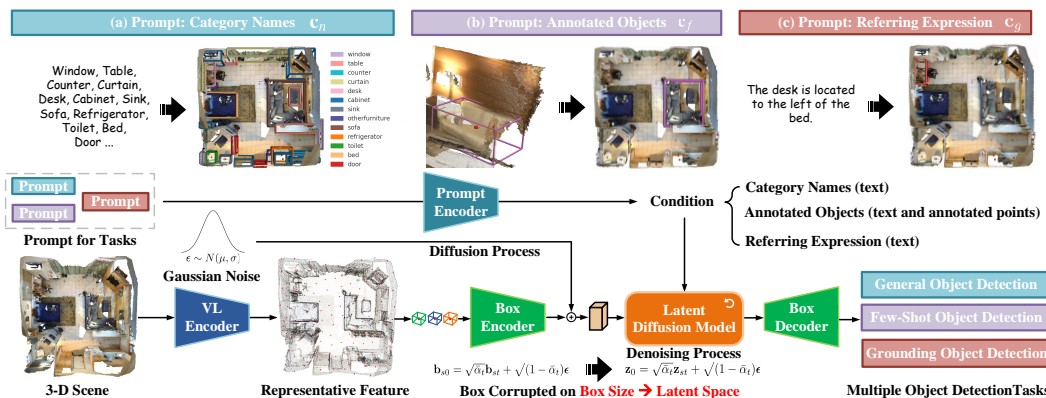

Figure 1: Promptable 3-D object localization via latent diffusion models: Conventional diffusion-based object detectors primarily apply stochastic perturbations directly to the bounding box coordinates. In contrast, our proposed approach perturbs the **entire bounding box representation within a learned latent space**, enabling a more structured and information-preserving diffusion process. This latent formulation, combined with **explicit conditioning mechanisms**, supports a controlled and adaptable diffusion framework. By incorporating these enhancements, the proposed method offers improved flexibility and robustness, facilitating seamless adaptation to a wide range of downstream tasks: *e.g.*, (a) general 3-D object detection, (b) few-shot detection, and (c) grounding-based detection.

computational efficiency and operational flexibility, echoing the inherent limitations encountered by DDPM [19] models in controlled image generation tasks.

Addressing these constraints, this research proposes an innovative diffusion-based object detection framework motivated by conditional latent diffusion models commonly utilized in generative modeling tasks. Specifically, as delineated in Figure 1, our approach draws conceptual parallels with DETR-based architectures [61, 75, 10]. Initially, object anchor features indicative of potential object presence within a scene are extracted and subsequently integrated with semantic textual embeddings corresponding to relevant object classes, thereby enriching feature representations. Subsequently, a variational autoencoder (box VAE) facilitates the mapping of object queries and their associated initial bounding-box coordinates into a latent embedding space. Leveraging the conditional latent diffusion paradigm, distinct conditioning criteria are systematically applied during the noise-to-box diffusion process, specifically tailored to downstream object detection scenarios, including general object detection, few-shot detection, and object grounding tasks. Distinguished from conventional approaches, the proposed methodology exhibits substantial adaptability, enabling seamless application across diverse detection objectives. The primary contributions of our method are as follows:

- Introduction of a conditional latent diffusion framework for improved adaptability and effectiveness in 3-D object detection over existing diffusion-based methods.
- Development of a versatile conditional latent diffusion paradigm capable of addressing multiple object detection tasks through adjustments of conditioning parameters alone.
- Empirical demonstration of competitive performance and superior flexibility in general 3-D object detection tasks and specialized downstream scenarios, notably 3-D few-shot detection and 3-D grounding detection.

## 2 Related work

### 2.1 Diffusion models in 3-D vision

Although initially designed for 2-D generation tasks, diffusion models have recently gained significant traction in 3-D applications. Common applications of diffusion models in 3-D include text to 3-D generation [46, 82, 76, 48] image to 3-D generation [36, 41, 44], 3-D editing and manipulation [59, 8], and novel view synthesis [37, 62, 21]. Due to their strong capability in modeling complex data distributions, several studies have attempted to leverage diffusion models for visual perception tasks, such as semantic segmentation [94] and language grounded classification [60], and human pose

estimation [14, 13, 20]. In this work, we focus on applying diffusion models to diverse 3-D object identification and localization tasks, which remain relatively underexplored.

## 2.2 Diffusion models for object detection

Diffusion models have achieved remarkable success in numerous visual perception downstream tasks. Chen *et al*. [7] proposed DiffusionDet, the first work to apply diffusion models to object detection. DiffusionDet frames the process as "**noise-to-box**", which starts with a fixed number of noisy box proposals and progressively refines them into the desired object boxes through a reverse denoising process. Building upon this framework, several works [93, 52, 30, 3] extend 2-D object detection to the 3-D domain and demonstrate promising results compared to previous anchor-based methods. Diffusion-SS3D [17] and Diff3DETR [10] leverage diffusion models to refine both bounding box proposals and class label distributions. This design, combined with a teacher-student framework, facilitates semi-supervised 3-D object detection. Despite achieving promising results, prior works directly predict box parameters, which limits their flexibility to generalize to other downstream tasks. In contrast, our work introduces a unified framework that encodes bounding boxes into a latent representation, enabling object identification and localization across diverse scenarios.

## 2.3 Few-shot 3-D object detection

The goal of few-shot 3-D object detection is to identify and localize objects in 3-D data by learning from base classes with abundant labeled data and generalizing them to new object categories with only a few labeled instances. Several existing approaches are based on the prototype learning paradigm [32, 90, 68]. Prototype learning extracts useful information from labeled data and utilizes the learned "prototype" to guide detection in unlabeled data. These methods [32, 90] are often built upon the VoteNet [50] architecture. Tang *et al*. [68] employ a VAE-based model to learn representative prototypes. Other works, such as Liu *et al*. [35], focus on few-shot 3-D object detection in outdoor scenes and autonomous driving scenarios. Meta-Det3D [84] addresses the few-shot 3-D object detection problem using meta-learning approach.

## 2.4 3-D visual grounding

3-D visual grounding aims to detect target objects that align with the given text description. A number of works formulate 3-D visual grounding as a segmentation task [91, 47, 87, 92, 11, 27, 80, 6, 31, 73, 67, 2, 28]. In contrast, our focus is on localizing objects that match the language query using bounding boxes. Methods for 3-D visual grounding can be roughly categorized into single-stage methods and two-stage methods. Single-stage methods [26, 42, 77, 51, 74, 16, 2] fuse text and vision features and directly output the predicted boxes based on the fused representations. In contrast, two-stage pipeline methods [5, 23, 81, 85, 12, 42, 26, 77, 95, 63, 51, 74, 57, 24, 88] first generate a fixed set of box proposals. In the second stage, each box candidate is matched with the text input to produce the final predictions. Our method extracts candidate object representations and encodes these features along with their corresponding coordinates into a structured latent space. A video stable diffusion model is then utilized to fuse this information with the provided text description. This fusion framework follows a structure similar to that of a two-stage pipeline. There are also works such as [22, 72, 86, 79, 95, 96] that leverage the power of large language models to tackle 3-D visual grounding tasks. Although these works do not specifically focus on 3-D visual grounding problems, our approach achieves stronger results under our evaluation protocol in most scenarios without relying on heavily pretrained large language models or complex fusion pipelines.

## 3 Method

Inspired by the "noise-to-box" paradigm employed in diffusion-based object detectors, we introduce a promptable latent diffusion detector designed to enhance the adaptability and precision of 3-D object detection. The architecture comprises three primary components: (1) A 3-D scene feature extraction module that encodes visual features, which are subsequently fused with semantic features to generate object anchor representations for the diffusion process (Section 3.2). (2) These object anchor representations, along with associated bounding box coordinates initialized with random perturbations, are processed through a box encoder module to obtain a compact latent representation

of the bounding box (Section 3.3). (3) The latent representation is then refined via a conditional latent diffusion model, which iteratively adjusts the bounding box parameters through a learned diffusion process conditioned on the latent space (Section 3.4). This structured approach ensures more flexible and accurate 3-D object localization by leveraging conditional priors, making it well-suited for applications that require precise and adaptable object detection in complex scenes.

## 3.1 Preliminaries

**3-D object detection** The task of 3-D object detection is fundamental in computer vision, involving the identification and precise localization of objects within a three-dimensional scene. Given a point cloud representation of a scene, $\{\mathbf{p}_i \in \mathbb{R}^3\}_{i=1}^n$, where $n$ denotes the number of points, along with a task-dependent promptable conditioning input $\mathbf{c}$, the goal is to predict a set of 3-D bounding boxes that accurately encapsulate target objects. Each bounding box is parameterized as $\mathbf{b} = (x, y, z, h, w, l, o_x, o_y, o_z) \in \mathbb{R}^9$, where $(x, y, z)$ represents the centroid of the box, $(h, w, l)$ specifies its spatial dimensions, and $(o_x, o_y, o_z)$ encodes its orientation. However, in practice, orientation data poses significant challenges due to inconsistencies between ground truth annotations and model predictions, as noted in [17, 83]. For instance, ScanNet [9] and SUN RGB-D [65] datasets either assign a default orientation of zero or contain inconsistent orientation data across scenes. Following the approach of [10, 83], we therefore consider only the centroids and sizes of the bounding boxes. The objective of generative object detection is to produce bounding boxes that effectively delineate individual object instances. In our formulation, by incorporating task-specific constraints defined by the promptable conditioning input $\mathbf{c}$, we explore generative approaches to produce bounding boxes adapted to various 3-D detection scenarios, including but not limited to 3-D object detection based on a few shots and grounding.

**Diffusion-based object detector** Recent advances in object detection have increasingly embraced the "noise-to-box" paradigm, as explored in both 2-D [7] and 3-D [17, 3, 10] settings. During training, the diffusion-based detection decoder $f_\theta$ estimates the clean bounding box $\mathbf{b}_0$ from a corrupted version $\mathbf{b}_t$, the visual features $\mathbf{x}$, and the corresponding timestep $t$. Unlike traditional diffusion models that predict the noise, this approach directly regresses $\mathbf{b}_0$ by minimizing the objective:

$$\mathcal{L}_\theta = \|f_\theta(\mathrm{RoI}(\mathbf{x}, \mathbf{b}_t), t) - \mathbf{b}_0\|^2, \tag{1}$$

where $\mathrm{RoI}(\cdot)$ denotes the region-of-interest alignment operation, used to extract relevant visual features. During inference, an initial set of randomly sampled noise boxes $\mathbf{b}_T$ is iteratively refined via the detection decoder and DDIM sampling steps [64], ultimately yielding the final predictions $\mathbf{b}_0$.

Unlike prior diffusion-based detection methods that directly regress bounding box coordinates, the proposed approach introduces a *conditional latent diffusion model* that predicts the noise in a latent space. This formulation is consistent with the prevalent practice in other diffusion-based generative modeling and offers improved flexibility for handling complex detection scenarios. Accordingly, the training objective is defined as:

$$\mathcal{L}_\theta = \|\boldsymbol{\epsilon}_\theta(\hat{\mathbf{b}}_t, \mathbf{c}, t) - \boldsymbol{\epsilon}\|^2, \tag{2}$$

where $\hat{\mathbf{b}}_t$ denotes the noisy latent representation obtained after applying $t$ forward diffusion steps to the encoded representation $\hat{\mathbf{b}} = \mathcal{E}(\mathbf{b}, \mathbf{o})$. Here, $\mathcal{E}$ denotes the Box VAE encoder, $\mathbf{o}$ represents the object anchor features, and $\mathbf{c}$ is the conditional input. While conceptually analogous to the object queries in DETR-style architectures, our anchors are enhanced with semantic information through cross-modal alignment between visual and textual embeddings, leading to more informative and context-aware representations. Further details are provided in Section 3.2 and Section 3.3.

## 3.2 Language-guided object anchor features

As illustrated in Figure 2 and aligned with prior works [10, 7], the proposed methodology adopts a DETR-based framework employing learned object anchors. For 3-D visual feature extraction, rather than exclusively utilizing conventional pretrained visual backbones such as PointNet++ [49] or PVCNN [39], we leverage a foundation model composed of two feature extractors, $f_v$ and $f_t$, to generate semantically coherent 3-D visual representations, thereby enriching the encoded features with enhanced contextual relevance. Given a point cloud representation of a scene, $\{\mathbf{p}_i \in \mathbb{R}^3\}_{i=1}^n$, where $n$ is the number of points, the visual features $\{\mathbf{z}_i^v\}_{i=1}^{n_1}$ are extracted via:

$$f_v : \mathbb{R}^3 \rightarrow \mathbb{R}^d \quad \text{and} \quad \{\mathbf{z}_i^v\}_{i=1}^{n_1} = f_v(\downarrow_{n_1}(\{\mathbf{p}_i\}_{i=1}^n)), \tag{3}$$

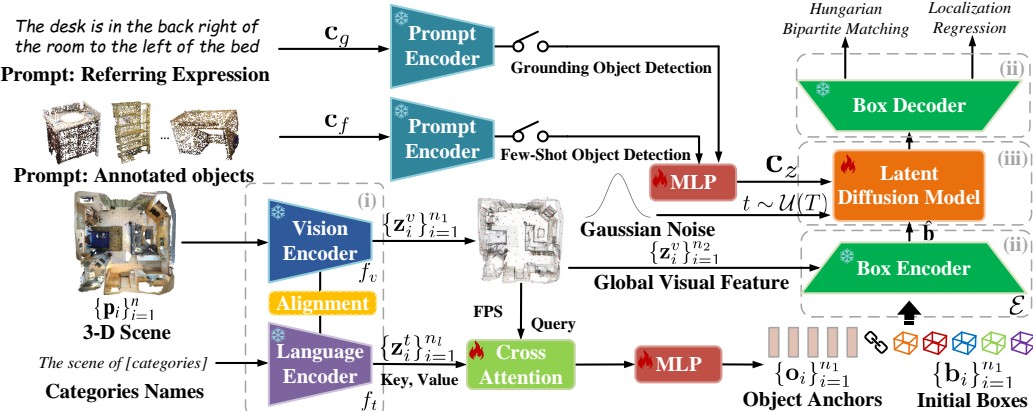

Figure 2: Model architecture. The proposed framework comprises three key components: (i) a vision-language foundation model that extracts object anchor features, serving as candidate object representations; (ii) a pretrained variational autoencoder (VAE) designed to encode bounding box coordinates into a structured latent space representation; and (iii) a latent-conditioned diffusion model that leverages conditioning features extracted by a dedicated encoder, facilitating adaptability across diverse downstream tasks by incorporating task-specific constraints within the latent space.

where $\downarrow_{n_1} (\cdot)$ denotes performing downsampling via Farthest Point Sampling (FPS) to yield a reduced set of $n_1$ points. To augment the object anchor features with semantic context beyond visual information, we introduce a cross-attention mechanism in which the extracted visual features $\{\mathbf{z}_i^v\}_{i=1}^{n_1}$ serve as queries, and the latent semantic representations $\{\mathbf{z}_i^t\}_{i=1}^{n_l}$, obtained from the text encoder $f_t$ (with token length $n_l$), act as keys and values. (See Figure 2.) This cross-attention produces the conditioned object query features $\{\mathbf{o}_i\}_{i=1}^{n_1}$ by:

$$Q^{(f)} = \varphi_q^{(f)}(\mathbf{z}^v), \quad K^{(f)} = \varphi_k^{(f)}(\mathbf{z}^t), \quad V^{(f)} = \varphi_v^{(f)}(\mathbf{z}^t) \in \mathbb{R}^d, \tag{4}$$

$$\mathbf{z} = \mathrm{MHCA}^{(f)}(Q^{(f)}, K^{(f)}, V^{(f)}) \in \mathbb{R}^d, \quad \mathbf{o} = \varphi_o^{(f)}(\mathbf{z}) \in \mathbb{R}^d, \tag{5}$$

where $\varphi_q^{(f)}(\cdot)$, $\varphi_k^{(f)}(\cdot)$ and $\varphi_v^{(f)}(\cdot)$ correspond to single-layer MLPs while $\varphi_o^{(f)}(\cdot)$ is a two-layer MLP to project the conditioned features effectively. The operator $\mathrm{MHCA}^{(f)}$ refers to a multi-head cross-attention module [71] that facilitates semantic alignment between modalities.

## 3.3 Box representation

Unlike existing diffusion-based detectors, which estimate the final bounding boxes solely based on their noisy versions, we propose leveraging a conditional latent diffusion processor to enhance flexibility. Similar to conventional latent diffusion approaches, the first step involves using a VAE module to project the input modality into a latent space. Specifically, we adopt a V-DETR [61]-like module as the VAE for the 3-D box representation, given its efficiency and high performance as demonstrated in [10]. As illustrated in Figure 3, object queries $\{\mathbf{o}_i\}_{i=1}^{n_1}$, along with their corresponding bounding boxes $\{\mathbf{b}_i\}_{i=1}^{n_1} = (x_i, y_i, z_i, h_i, w_i, l_i)$ and the global visual features of the scene $\{\mathbf{z}_i^v\}_{i=1}^{n_2}$ ($n_2$ is not equal to $n_1$), are passed through a box encoder to extract the latent box representation. The first component of the encoder is a variant of the multi-head self-attention (MHSA) module (including attention and residual operators) applied to object anchors:

$$\mathbf{b}^z = \psi_b(\mathbf{b}), \quad Q^s = \psi_q(\mathbf{b}^z + \mathbf{o}), \quad K^s = \psi_k(\mathbf{b}^z + \mathbf{o}), \quad V^s = \psi_v(\mathbf{o}) \in \mathbb{R}^d, \tag{6}$$

$$\mathbf{b}^s = \mathrm{MHSA}(Q^s, K^s, V^s) \in \mathbb{R}^d, \tag{7}$$

where $\psi_b(\cdot)$, $\psi_q(\cdot)$, $\psi_k(\cdot)$, and $\psi_v(\cdot)$ represent single-layer MLPs. The second component is a cross-attention module that takes the output $\mathbf{b}^s$ from the self-attention module and the visual features of the scene $\{\mathbf{z}_i^v\}_{i=1}^{n_2}$ as inputs:

$$Q^c = \varphi_q(\mathbf{b}^z + \mathbf{b}^s), \quad K^c = \varphi_k(\mathbf{z}^v + RPE), \quad V^c = \varphi_v(\mathbf{z}^v) \in \mathbb{R}^d, \tag{8}$$

$$\hat{\mathbf{b}} = \mathrm{MHCA}(Q^c, K^c, V^c) \in \mathbb{R}^d, \tag{9}$$

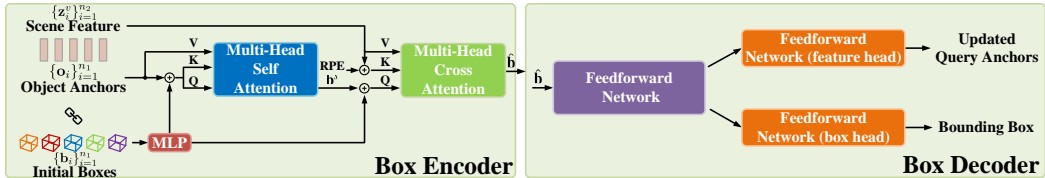

Figure 3: The Box VAE takes as input object anchor features, 3-D spatial coordinates, and scene representations. Initial box centers $(x_i, y_i, z_i)$ are determined via FPS on anchor positions, while box sizes $(h_i, w_i, l_i)$ are initialized using the dataset's normalized average dimensions. In the box encoder, self-attention is applied with queries and keys formed by summing object anchor features and coordinate embeddings, and values taken as the anchor features. A cross-attention module further refines this representation by conditioning on scene-level features, enhancing spatial-contextual awareness. The box decoder employs two feedforward networks: one predicts final bounding boxes, while the other updates query anchor features, enabling robust refinement and precise localization.

where $\varphi_q(\cdot)$, $\varphi_k(\cdot)$, and $\varphi_v(\cdot)$ are single-layer MLPs, while $RPE$ represents the relative position encoding. In the original V-DETR, the output of the cross-attention module, $\hat{\mathbf{b}}$, is passed through three independent FFN heads to predict the class, rotation angle, and bounding box. In our formulation, we treat the two attention modules as the encoder of the box VAE and the FFN heads as the decoder. Since our approach requires utilizing $\hat{\mathbf{b}}$ as the latent representation of the bounding boxes for the latent diffusion process, we modify the architecture accordingly.

### 3.4 Box refinement via conditional latent diffusion

Latent diffusion models (LDMs) [58] project the original data $\mathbf{x}$ into a latent space $\mathbf{z}$ using an encoder tailored to the data modality—e.g., VQ-VAE [70, 53] for images or audio VAE [34] for audio—prior to applying the diffusion process. When combined with classifier-free guidance [18], LDMs enable conditional generation. The model is trained by minimizing the following objective:

$$\mathcal{L}_\theta = \|\epsilon_\theta(\mathbf{z}_t, \mathbf{c}_z, t) - \epsilon\|^2. \tag{10}$$

Here, $\epsilon_\theta(\mathbf{z}_t, \mathbf{c}_z, t)$ incorporates a cross-attention mechanism [71], allowing the conditional embedding $\mathbf{c}_z$ to influence the latent variable $\mathbf{z}$. The conditional score estimation is then defined as:

$$\hat{\epsilon}_\theta(\mathbf{z}_t|\mathbf{c}_z) = (1 + \omega)\epsilon_\theta(\mathbf{z}_t, \mathbf{c}_z, t) - \omega\epsilon_\theta(\mathbf{z}_t, \varnothing, t) \tag{11}$$

where $\omega$ modulates the strength of classifier-free guidance, and $\varnothing$ denotes the unconditional embedding. To integrate the latent representations of bounding boxes $\hat{\mathbf{b}}$ into the diffusion process, we adapt the conditioning mechanism defined in Equations (10) and (11) by substituting $\mathbf{z}_t$ with $\hat{\mathbf{b}}_t$, the noisy latent representation of the bounding boxes. This conditional latent diffusion process enables iterative refinement of box features through dedicated decoder heads, which generate updated object anchors and bounding boxes conditioned on contextual information. In our implementation, the model is initialized using a pretrained video-based LDM from Stable Diffusion, providing a strong prior for temporal and spatial coherence in the latent space.

**Model training and inference**  As depicted in Figure 2, the proposed framework jointly optimizes a cross-attention-based object anchor generator and a conditional latent diffusion model. The overall training objective integrates a latent-space denoising loss with detection losses applied to the decoded bounding boxes. Specifically, the total loss is defined as:

$$\mathcal{L}_{\text{total}} = \lambda_{\text{diff}} \cdot \mathcal{L}_{\text{diff}} + \lambda_{\text{det}} \cdot \mathcal{L}_{\text{det}}, \tag{12}$$

where $\mathcal{L}_{\text{diff}}$ denotes the mean squared error between the predicted and true noise in the latent space, as defined in Equation (2), and $\mathcal{L}_{\text{det}}$ includes classification and regression losses computed on the decoded bounding boxes. We adopt a DETR-style pipeline, incorporating Hungarian bipartite matching and non-maximum suppression (NMS). To further enhance detection accuracy and robustness, focal loss [33] and asymmetric classification losses [66, 56] are additionally employed. The weighting coefficients $\lambda_{\text{diff}}$ and $\lambda_{\text{det}}$ are dynamically adjusted throughout training, as detailed in Section 4.

At inference time, each bounding box is initialized with a noisy latent representation $\hat{\mathbf{b}}_T$, corresponding to the final timestep of the forward diffusion process. This representation is iteratively refined via DDIM sampling, guided by the detection decoder and conditioned on object anchor features and task-specific prompts, ultimately producing the final bounding box predictions $\hat{\mathbf{b}}_0$.

**Promptable 3-D vision tasks**    Owing to its conditional and modular design, the proposed framework supports a wide range of 3-D object detection tasks, including general detection, few-shot detection, and grounding. Each task is described as follows:

- General 3-D object detection: The objective is to detect all objects present in a 3-D scene. During training, the conditional prompt $\mathbf{c}_n$ is expressed as a textual description: "*The [class name1, class name2, ... class nameN] objects in the 3D scene.*" This aligns semantic cues with visual features to enhance representation learning. As class names are already used in anchor generation, their inclusion in the prompt is optional and omitting them incurs only a minor performance drop.

- Few-Shot 3-D object detection (FS3D): Following prior work [90, 68], the full class set $C$ is divided into base classes $C_{\text{base}}$ with ample labels and novel classes $C_{\text{novel}}$ with limited samples, where $C_{\text{base}} \cap C_{\text{novel}} = \varnothing$. To address this setting, we adopt an episodic training strategy and introduce architectural modifications to support FS3D (detailed in the supplementary material). The query input consists of a 3-D scene, while the support set is processed by a 3-D encoder to extract visual exemplars. These exemplars, combined with textual class names, form the conditioning input $\mathbf{c}_f$.

- Grounding 3-D object detection: This task aims to localize only the objects referred to by a natural language query, rather than detecting all instances in the scene. Referring expressions are more flexible than fixed category names and may include spatial, relational, or attribute-based cues. These expressions serve as the conditional input $\mathbf{c}_g$ to the latent diffusion process. To handle grounding scenarios in which a single query may correspond to multiple target objects, as in Multi3DRefer, we extend the inference pipeline with non-maximum suppression and top-K filtering. This allows the DETR-style framework to return multiple high-confidence predictions per query. The query-conditioned diffusion process then refines candidate boxes for accurate localization.

## 4    Experiments

In this section, we present experimental results aimed at demonstrating the effectiveness of the proposed method.

**Training and loss functions**    The training process involves two primary stages: (1) training the diffusion model with a frozen VAE, and (2) fine-tuning the decoder. In the first stage, contrary to previous works [10, 17] that typically use mean values of 0.25 for size noise and $1/n_{\text{class}}$ for label sampling, we set the latent noise mean to 0.1, empirically demonstrating improved performance. Optimization employs the Adam optimizer with an initial learning rate of $5 \times 10^{-4}$, a cosine annealing schedule featuring a 500-step linear warm-up, and a minimum learning rate of $1 \times 10^{-6}$ to ensure stable convergence.

During the diffusion stage, the loss function follows Equation (2), combining latent-space denoising loss $\mathcal{L}_{\text{diff}}$ and auxiliary detection loss $\mathcal{L}_{\text{det}}$, as defined in the Method section. To reuse pretrained video LDM weights, we add a lightweight MLP adapter between the V-DETR box-latent and the diffusion backbone. Concretely, given $\hat{\mathbf{b}} \in \mathbb{R}^{N \times d}$, a two-layer MLP projects it to a 4-channel canvas $\hat{\mathbf{b}}' \in \mathbb{R}^{4 \times H_b \times W_b}$ with $H_b \times W_b = N$ on which the video-LDM U-Net operates. Its output is mapped back via another two-layer MLP to $\mathbb{R}^{N \times d}$ and decoded to boxes. We compute the diffusion loss $\mathcal{L}_{\text{diff}}$ on the noised canvas $\hat{\mathbf{b}}'_t$ and the detection loss $\mathcal{L}_{\text{det}}$ after the inverse projection and the box decoder. This matches the 4-channel latent convention in Stable Video Diffusion. The model trains for 18K iterations with a batch size of 8, accumulating gradients over 16 steps. The decoder remains frozen at this stage, with $\mathcal{L}$det indirectly guiding latent refinement. In the second stage, we unfreeze the VAE decoder, keeping the encoder fixed, and fine-tune the decoder with detection losses, using a reduced learning rate and regularization to maintain pretrained decoder stability. To balance training objectives, the loss coefficients initially set as $\lambda_{\text{diff}} = 1.0$, $\lambda_{\text{det}} = 0.2$ gradually adjust to $\lambda_{\text{diff}} = 0.5$,

Table 1: General object detection on SUN RGB-D and ScanNetV2 datasets.

| Method | SUN RGB-D [9] | | ScanNetV2 [65] | |
|---|---|---|---|---|
| | mAP@25 | mAP@50 | mAP@25 | mAP@50 |
| VoteNet [50] | 57.9 | 29.3 | 57.8 | 36.0 |
| 3DETR [45] | 59.1 | 32.7 | 65.0 | 47.0 |
| Group-Free [40] | 63.0 | 45.2 | 69.1 | 52.8 |
| Uni3DETR [75] | 67.0 | 50.3 | 71.7 | 58.3 |
| V-DETR [61] | **67.5** | **50.4** | **77.4** | **65.0** |
| *Diffusion-based detector* | | | | |
| Diffusion-SS3D [17] | - | - | 64.1 | 43.2 |
| Diff3DETR [10] | - | - | 65.7 | 44.9 |
| CatFree3D [3] | - | - / 52.0† | - | - |
| Ours | 67.4 | 50.2 / 54.5† | 72.8 | 60.3 |

† : Training on 31 categories (including background) and testing on the other 7.

Table 2: Few-shot object detection on FS-SUNRGBD. Bold texts denote the best results on each scenario.

| Method | FS-SUNRGBD [90] | | | | | |
|---|---|---|---|---|---|---|
| | 1-shot | | 3-shot | | 5-shot | |
| | mAP@25 | mAP@50 | mAP@25 | mAP@50 | mAP@25 | mAP@50 |
| VoteNet [50] | 5.46 | 0.22 | 13.73 | 2.20 | 22.99 | 5.90 |
| GeneralizedFS3D [35] | 6.81 | 1.58 | 17.52 | 4.69 | 22.84 | 6.76 |
| PointContrast-VoteNet [78] | 7.03 | 1.17 | 20.32 | 4.19 | 21.03 | 6.71 |
| Fractal-VoteNet [75] | 7.54 | 1.39 | 21.08 | 4.25 | 22.01 | 6.77 |
| Meta-Det3D [84] | 6.77 | 0.73 | 15.37 | 2.99 | 24.22 | 5.68 |
| Prototypical-VoteNet [90] | 12.39 | 1.52 | 21.51 | 6.13 | 29.95 | 8.16 |
| Prototypical-VAE [68] | 14.36 | 2.42 | 27.70 | 8.73 | 33.21 | 13.98 |
| Ours | **20.69** | **6.52** | **34.72** | **13.52** | **40.52** | **20.25** |

Table 3: Few-shot object detection on FS-ScanNet. Bold texts denote the best results on each scenario.

| Method | FS-ScanNet [90] | | | | | | | | | | | |
|---|---|---|---|---|---|---|---|---|---|---|---|---|
| | Split-1 | | | | | | Split-2 | | | | | |
| | 1-shot | | 3-shot | | 5-shot | | 1-shot | | 3-shot | | 5-shot | |
| | mAP@25 | mAP@50 | mAP@25 | mAP@50 | mAP@25 | mAP@50 | mAP@25 | mAP@50 | mAP@25 | mAP@50 | mAP@25 | mAP@50 |
| VoteNet [50] | 11.72 | 8.02 | 21.13 | 9.57 | 28.63 | 15.69 | 8.79 | 1.71 | 18.19 | 5.52 | 22.68 | 11.64 |
| GeneralizedFS3D [35] | 12.03 | 8.19 | 24.90 | 10.26 | 29.29 | 16.67 | 9.19 | 1.87 | 19.41 | 6.80 | 25.18 | 12.74 |
| PointContrast-VoteNet [78] | 12.59 | 8.52 | 25.83 | 11.16 | 25.65 | 15.49 | 9.55 | 1.97 | 18.44 | 5.23 | 20.06 | 10.19 |
| Fractal-VoteNet [75] | 11.81 | 7.57 | 21.38 | 10.11 | 24.66 | 14.73 | 9.16 | 1.68 | 15.65 | 4.88 | 20.35 | 10.26 |
| Meta-Det3D [84] | 10.28 | 4.03 | 23.42 | 10.64 | 25.65 | 13.88 | 5.21 | 1.32 | 15.44 | 4.37 | 22.13 | 7.09 |
| Prototypical-VoteNet [90] | 15.34 | 8.25 | 31.25 | 16.01 | 32.25 | 19.52 | 11.01 | 2.21 | 21.14 | 8.39 | 28.52 | 12.35 |
| Prototypical-VAE [68] | 16.00 | 10.22 | 31.60 | 19.37 | 32.84 | 22.39 | 12.66 | 4.15 | 21.27 | 10.09 | 31.70 | 14.43 |
| Ours | **20.34** | **13.64** | **36.75** | **24.42** | **37.45** | **26.54** | **17.23** | **6.37** | **25.63** | **13.53** | **41.36** | **19.75** |

$\lambda_{\det} = 1.0$. We present the training and inference process in Algorithm 1 and 2. All experiments utilize eight NVIDIA RTX A6000 Ada GPUs.

## 4.1 General 3-D object detection

**Datasets and evaluation metrics** We evaluate the proposed method on two standard indoor benchmarks: SUN RGB-D and ScanNet. SUN RGB-D includes 5,285 training scenes along with corresponding validation scenes, while ScanNet comprises 1,201 training and 312 validation scenes reconstructed from 2.5 million RGB-D frames. Following prior works [17, 10, 50, 61], we evaluate on the 10 most common object classes for SUN RGB-D and 18 semantic classes for ScanNet. Performance is measured using mean Average Precision (mAP) at IoU thresholds of 0.25 and 0.5. All results are averaged over three random splits, and we report both the mean and standard deviation.

**Results** Table 1 presents the results on the general 3-D object detection task. The proposed method achieves approximately a 5% improvement in mAP on ScanNetV2 compared to other diffusion-based detectors, and it demonstrates competitive performance on SUN RGB-D compared to state-of-the-art approaches. Note that while V-DETR uses a three-layer cascade structure, we use only a single layer. Additionally, for fair comparison, we also report results under CaTFree3D's experimental setting.

## 4.2 Few-shot 3-D object detection

**Datasets and evaluation metric** We test our method on two few-shot 3-D object detection benchmarks: FS-SUNRGBD and FS-ScanNet [90]. FS-SUNRGBD contains 5,000 point-cloud scenes spanning 10 object categories, while FS-ScanNet includes 1,513 scenes across 18 categories. The base/novel splits are 6/4 for FS-SUNRGBD and 12/6 for FS-ScanNet. Following standard protocols [68, 90], we report mAP at IoU thresholds of 0.25 and 0.5 under varying shot settings.

**Results** As described in Section 3.4, following [68, 90], we adopt an episodic training strategy. To adapt the scenario, we leverage support sets consisting of annotated point cloud features and text prompts as conditional inputs to the latent diffusion model. The results on FS-SUNRGBD and FS-ScanNet under 1-shot, 3-shot, and 5-shot settings are summarized in Tables 2 and 3. Across all settings, the proposed method achieves approximately a 4% improvement in mAP, demonstrating its superior performance and strong generalization capability in the few-shot regime.

## 4.3 Grounding 3-D object detection

**Datasets and evaluation metrics** We evaluate the proposed method on three benchmarks for 3-D visual grounding: ScanRefer [5], Multi3DRefer [89], and ViGiL3D [72]. Specifically, ScanRefer

Table 4: Grounding object detection on ScanRefer, Multi3DRefer, and ViGiL3D.

| Method | ScanRefer [5] | | Multi3DRefer [89] | | ViGiL3D [72] | |
|---|---|---|---|---|---|---|
| | Acc@25 | Acc@50 | F1@25 | F1@50 | Acc@25 | Acc@50 |
| ScanRefer [5] | 37.3 | 24.3 | - | - | - | - |
| Multi3DRefer [89] | 51.9 | 44.7 | 42.8 | 38.4 | - | - |
| ConcreteNet [69] | 46.5 | 46.5 | - | - | - | - |
| D-LISA [88] | - | 46.9 | - | 51.2 | - | - |
| Chat-Scene [22] | 55.5 | 50.2 | 57.1 | 52.4 | 11.0† | 9.7† |
| PQ3D [96] | 57.0 | 51.2 | - | 50.1 | 10.8 | 10.8 |
| **Ours** | **59.5** | **52.7** | **59.4** | **53.8** | **15.7** | **13.3** |

† : Results produced by our evaluations with the provided code.

Table 5: Zero-shot grounding object detection on the open-vocabulary benchmark OpenLex3D.

| Method | OpenLex3D [28] | | | | | |
|---|---|---|---|---|---|---|
| | Replica [5] | | ScanNet++ [89] | | HM3D [72] | |
| | Acc@25 | Acc@50 | F1@25 | F1@50 | Acc@25 | Acc@50 |
| OpenMask3D [67] | **21.5** | 15.1 | 9.8 | 4.2 | 8.2 | 5.3 |
| ConceptGraphs [15] | 19.4 | 16.2 | **11.5** | **5.4** | 9.4 | 6.6 |
| **Ours** | 19.5 | **17.9** | 11.3 | **5.4** | **9.9** | **7.6** |

focuses on single-object grounding, Multi3DRefer involves multi-object grounding per query, and ViGiL3D serves as a diagnostic benchmark featuring a mixture of single, multiple, and no-target queries. Following their respective evaluation protocols, we report Acc@25 and Acc@50 for ScanRefer, and F1@25 and F1@50 for both Multi3DRefer and ViGiL3D. During training, we adopt a DETR-style matching strategy using Hungarian bipartite assignment. At inference time, we retain high-confidence predictions via non-maximum suppression (NMS), which is consistently applied across all tasks. Notably, while Hungarian matching is suited for single-object settings such as ScanRefer, it cannot directly support multi-object queries. Therefore, for Multi3DRefer and ViGiL3D, we follow prior work [89] and apply a multi-match evaluation strategy based on IoU thresholds and label agreement to compute precision, recall, and F1 scores. We further evaluate our method in a zero-shot setting on the open-vocabulary benchmark OpenLex3D [28]. Since the dataset provides only semantic segmentation masks and synonym lists, we construct axis-aligned bounding boxes (AABB) from the segmentation results and use the first synonym in each list as the reference label. Accuracy is reported at IoU thresholds of 0.25 and 0.5.

**Results** Table 4 shows that our method achieves strong performance across all three grounding benchmarks, demonstrating robustness in both single- and multi-object scenarios. In Table 5, it also achieves competitive results on OpenLex3D under a zero-shot setting, highlighting strong generalization in open-vocabulary 3D understanding. While OpenMask3D and ConceptGraphs report slightly higher scores on some subsets, they rely on complex multi-stage pipelines involving multi-view fusion, explicit mask decoding, or large language models like CLIP, LLaVA, and GPT-4. In contrast, our approach leverages CLIP2Point for visual-language alignment and achieves comparable or better performance without external LLMs, handcrafted scene graphs, or post-processing. This underscores the efficiency of promptable latent diffusion for open-vocabulary 3-D object grounding.

### 4.4 Ablation study and discussion

As shown in Figure 4, we perform three ablation studies across eight 3-D benchmarks to evaluate three key components of our method.

**Language-guided object anchors** Replacing the cross-attended anchors with purely visual features leads to consistent performance drops, particularly on open-vocabulary benchmarks such as ViGiL3D (-2.9) and OpenLex3D (-2.7), where semantic alignment is crucial. The degradation on Multi3DRefer (-4.4) and ScanRefer (-5.2) further highlights the importance of language-guided anchoring for accurate grounding.

**Latent diffusion refinement** Substituting our diffusion module with direct regression significantly degrades few-shot performance, with FS-ScanNet dropping from 13.64 to 8.24 (-5.4) and FS-SUNRGBD from 6.52 to 4.32 (-2.2). Grounding accuracy also declines, *e.g.*, ViGiL3D: (-4.1), Multi3DRefer: (-6.4), indicating the value of iterative refinement under limited data.

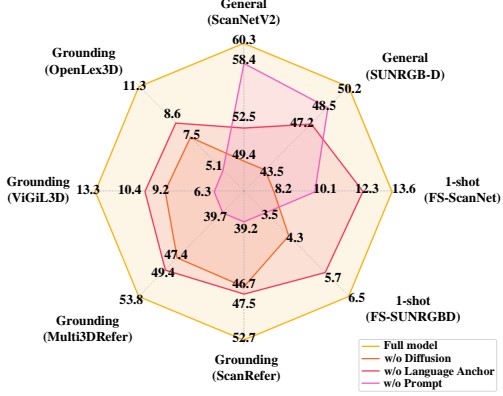

Figure 4: Ablation study on eight 3-D detection and grounding benchmarks, comparing the full model with versions without latent diffusion, language-guided anchors, or prompt conditioning. Each component contributes significantly, especially in open-vocabulary grounding tasks.

Table 6: Summary of key characteristics of diffusion-based approaches versus the proposed method.

| Method | Prompt modality | Detection task | Representative score |
|--------|-----------------|----------------|----------------------|
| DiffusionDet [7] | – | 2-D detection | COCO: 45.8 mAP@50 |
| GroundingDINO [38] | text | 2-D detection | LVIS : 32.5 mAP@50 |
| Diffusion-SS3D [17] | – | 3-D detection | ScanNetV2: 43.2 mAP@50 |
| Diff3DETR [10] | – | 3-D detection | ScanNetV2: 44.9 mAP@50 |
| **Ours** | **text, image** | **3-D detection, Few-shot, Grounding** | **8 datasets w/ prompt + 3% (↑)** |

**Promptable conditioning** Removing prompt conditioning causes the most severe degradation across grounding tasks. Performance on ScanRefer drops from 52.7 to 39.2 (-13.5), Multi3DRefer from 53.8 to 39.7 (-14.1), and ViGiL3D from 13.3 to 6.3 (-7.0). Even in one-shot settings, *e.g.*, FS-SUNRGBD: (-3.1), performance declines, confirming the necessity of prompts for semantic guidance, particularly in ambiguous or data-scarce conditions.

Overall, the ablation results are consistent with our design goals: promptable conditioning is critical for language-driven tasks, diffusion enables robust learning in low-data regimes, and language-guided anchors improve semantic grounding, particularly in open-vocabulary settings.

**Core contributions of the proposed method** As shown in Table 6, our diffusion-based detector is built upon the concept of a noise-to-box, reflecting the recent advancements of diffusion models across computer vision, including both discriminative tasks and dense prediction scenarios. Recent methods such as Marigold [29] for depth estimation further highlight the flexibility of diffusion-based models. However, existing diffusion-based detectors, such as DiffusionDet [7] and Diffusion-SS3D [17], typically diffuse only raw box coordinates. This design restricts their ability to incorporate arbitrary language inputs or exemplar-based prompts directly, thereby limiting their generalization capacity. Our method addresses these limitations by leveraging latent diffusion models, which can seamlessly integrate diverse multimodal inputs. By embedding an aligned foundational model into the diffusion process, our approach achieves a high degree of flexibility and adaptability across a wide range of tasks and modalities. Unlike standard detectors, our framework natively supports various input types, allowing for flexible and precise prediction control via prompt-based conditioning. This feature significantly broadens the practical utility of diffusion-based detection frameworks. The main novelty of our method is the integration of prompt conditioning into the diffusion-based detector through a noise-to-box paradigm, as emphasized in the Introduction and Conclusion. Our approach draws inspiration from recent findings, such as the work "Multimodality Helps Few-shot 3-D Point Cloud Semantic Segmentation", which demonstrates the benefits of leveraging multiple modalities [1]. We specifically employ a latent diffusion model due to its intrinsic ability to fuse multimodal information effectively. While existing diffusion-based detectors are usually restricted to a single task or modality (*e.g.*, DiffusionDet is limited to 2-D detection, Diffusion-SS3D and Diff3DETR [10] only address closed-set 3D detection without prompt conditioning, and GroundingDINO [38], although promptable, is confined to 2D detection tasks), our proposed framework overcomes these barriers. By utilizing a latent box VAE, our method enables a single trained model to flexibly support closed-set detection, few-shot adaptation, and language-driven 3D detection, all by simply modifying the prompts (*e.g.*, class names, visual exemplars, or natural language descriptions).

## 5   Conclusion

We propose a unified and flexible framework for 3-D object detection that reformulates the diffusion process as a conditioned noise-to-box transformation. Unlike prior diffusion-based detectors, our method introduces a conditional latent diffusion model that enables promptable box generation within a latent space. By incorporating task-specific prompts, our approach seamlessly adapts to diverse 3-D detection settings, including general, few-shot, and grounding-based scenarios. To enhance detection quality, we adopt a DETR-style architecture to generate semantically rich object anchors, which serve as informative inputs to the diffusion process. Extensive experiments across various benchmarks demonstrate the versatility and strong performance of our method, highlighting both its generalization capabilities and the potential of diffusion-based modeling in 3-D object detection.

**Acknowledgements.** This work was supported in part by NSTC grants 113-2221-E-001-010-MY3 and 113-2634-F-007-002 of Taiwan. We thank National Center for High-performance Computing for providing computing resources.

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

# A Additional implementation details

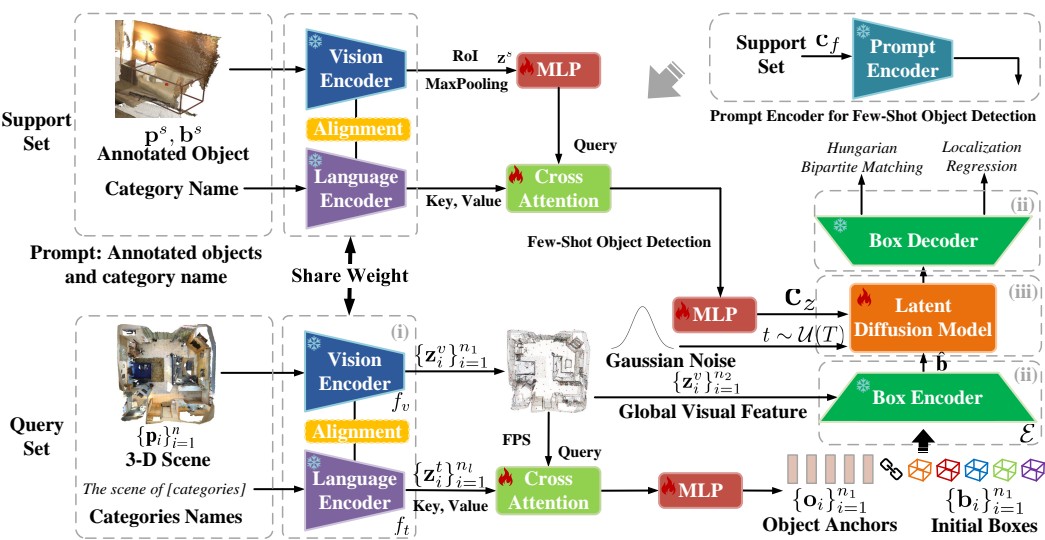

Figure 5: Model architecture for the few-shot setting.

**Architecture for few-Shot object detection** We adopt an episodic training paradigm to address few-shot object detection and introduce architectural modifications tailored to the FS3D setting. Unlike general or grounding-based 3-D detection, our FS3D framework employs a dual-branch architecture that separately processes a query point cloud and a small set of annotated support scenes. Each query input is a complete 3-D scene represented as a point cloud $\{\mathbf{p}_i \in \mathbb{R}^3\}_{i=1}^n$, which is processed by a backbone network to produce per-point features $\{\mathbf{z}_i \in \mathbb{R}^d\}_{i=1}^{n_1}$, where $d$ denotes the feature dimension.

For the support set, each support sample is also a full 3-D scene ($\{\mathbf{p}_i^s \in \mathbb{R}^3\}_{i=1}^n$. However, only the points enclosed within ground-truth bounding boxes are used to represent the relevant object instances. Specifically, for each bounding box ($\mathbf{b}^s = (x, y, z, h, w, l)$), we extract the subset of points within the region of interest (RoI) and obtain their corresponding features via the backbone network. These features are then aggregated using max pooling to produce an instance-level representation of the object:

$$\mathbf{z}^s = \mathrm{MaxPool}(f_v^s(\mathrm{RoI}(\mathbf{p}^s, \mathbf{b}^s))). \tag{13}$$

In the FS3D prompt encoder, these instance-level features are passed through an additional MLP and a cross-attention module to produce class-aware and instance-aware latent representations. These latents are subsequently used to condition the diffusion-based detector. This design differs from prior FS3D approaches that typically fuse support and query features via cross-attention at detection time. Instead, the proposed method treats the support set as an explicit conditioning signal for the generative latent process, promoting flexibility and compatibility with generalizable multimodal prompts.

**Language-guided object anchor features** For object anchor generation, we employ two vision-language pretrained models, CLIP2Point [25] and Uni3D [92], whose vision encoders align with the CLIP text encoder, facilitating seamless multimodal integration. As detailed in Section 3.2, we generate $n_1 = 128$ object anchors $\{\mathbf{o}_i\}_{i=1}^{n_1}$, initializing the corresponding bounding boxes $\{\mathbf{b}_i\}_{i=1}^{n_1}$ using Farthest Point Sampling (FPS) for centroids. The dimensions of the box are initialized with the average dimensions of the dataset or randomly within a normalized range $[0, 1]$.

**Box representation** For the Box VAE, we utilize a pretrained one-layer V-DETR module [61], omitting the ordinal prediction branch and adapting the global feature from the visual encoder ($n_2 = 1024$). Note that the original V-DETR employs a three-stage cascade architecture, whereas our approach leverages a simplified, single-layer adaptation.

**Algorithm 1:** Training

```python
def train(pc, gt_b, gt_l, clsn, cond, T):
  # Extract 3-D scene features
  pts, zv = foundation.encoder.v(pc)
  zt = foundation.encoder.t(clsn)

  # Compute conditional embeddings
  cz = prompt.encoder(cond)

  # Generate object anchor features
  bo = cross_attention(zv, zt)

  # Initialize bounding boxes
  bb_init = init_boxes(bo)

  # Encode to latent space
  bb_latent = box_vae.encoder(bb_init, bo)

  # Sample random diffusion timestep
  t = randint(1, T)

  # Add noise to latent
  eps = normal(mean=0, std=1)
  bb_noisy = corrupt(bb_latent, t, eps)

  # Predict noise with diffusion model
  eps_pred = ldm(bb_noisy, cz, t)

  # Compute diffusion loss
  L_diff = mse(eps_pred, eps)

  # Decode latent to boxes
  bb_pred = box_vae.decoder(bb_latent)

  # Compute detection loss
  L_det = detection_loss(bb_pred, gt_b, gt_l)
  loss = L_diff + L_det
  update(model, loss)

  return loss
```

corrupt(x, t, eps):sqrt(     alpha_cumprod(t)) * x +
                  sqrt(1 - alpha_cumprod(t)) * eps
alpha_cumprod(t):   $\prod_{i=1}^{t} \alpha_i$

**Algorithm 2:** Inference

```python
def inference(pc, clsn, cond, T, steps):
  # Extract 3-D scene features
  pts, zv = foundation.encoder.v(pc)
  zt = foundation.encoder.t(clsn)

  # Compute conditional embeddings
  cz = prompt.encoder(cond)

  # Generate object anchor features
  bo = cross_attention(zv, zt)

  # Initialize noisy latent boxes
  bb_init = init_boxes(bo)

  # Encode to latent space
  bb_latent = box_vae.encoder(bb_init, bo)

  # Add random noise in latent domain
  eps = normal(mean=0, std=1,
  size=bb_latent.shape)
  bb_noisy = corrupt(bb_latent, t=T, noise=eps)

  # Prepare sampling schedule
  time_points = linspace(-1, T, steps)
  times = reversed(time_points)
  pairs = list(zip(times[:-1], times[1:]))

  for t_cur, t_next in pairs:
    # Predict noise at current step
    eps_pred = ldm(bb_noisy, cz, t_cur)

    # DDIM update of latent
    bb_noisy = ddim(bb_noisy, eps_pred, t_cur,
  t_next)

  # Decode final latent to boxes
  bb_final = box_vae.decoder(bb_noisy)

  return bb_final
```

linspace:generate evenly spaced values

**Box Refinement via Conditional Latent Diffusion**  We use the DDIM [64] noise scheduler with a maximum of 1000 timesteps, initializing the latent diffusion model from pre-trained stable video diffusion weights [4]. Classifier-free guidance [18] is applied with a guidance scale of 3.5. For the prompt encoder, we adopt the CLIP text encoder for general and grounding object detection tasks, benefiting from its well-established alignment and widespread use in latent diffusion models.

# B  More experimental results

## B.1  Performance consistency and stability

As mentioned above, due to space constraints, only the mean performance is reported in the main paper. To support the reported main results in the paper, we include the performance over three independent runs with different random seeds for each benchmark. As shown in Table 7, the proposed method achieves consistent results with low variance on general, few shot, and grounding 3-D object detection tasks. This demonstrates the stability and robustness of the approach despite the diverse task settings.

## B.2  Ablation study

In addition to the core components discussed in the main paper, we further investigate the impact of different box initialization strategies. Specifically, we compare random initialization with a strategy that uses the average object size computed from the training set as a prior. As shown in Table 8, using average size initialization consistently improves performance across all three benchmarks:

Table 7: Performance of the proposed method across different 3-D detection and grounding benchmarks. We report the mean and standard deviation over three independent runs with different random seeds, illustrating the consistency and robustness of the method.

| Dataset | mean Average Precision | | Accuracy | | F1-Score | |
|---|---|---|---|---|---|---|
| | mAP@25 | mAP@50 | Acc@25 | Acc@50 | F1@25 | F1@50 |
| *General 3-D object detection* | | | | | | |
| SUN RGB-D [9] | $67.4 \pm 0.9$ | $50.2 \pm 0.5$ | - | - | - | - |
| ScanNetV2 [65] | $72.8 \pm 1.3$ | $60.3 \pm 0.7$ | - | - | - | - |
| *Few-shot 3-D object detection* | | | | | | |
| FS-SUNRGBD (1-shot) [90] | $20.69 \pm 1.32$ | $6.52 \pm 0.62$ | - | - | - | - |
| FS-SUNRGBD (3-shot) [90] | $34.72 \pm 1.71$ | $13.52 \pm 0.53$ | - | - | - | - |
| FS-SUNRGBD (5-shot) [90] | $40.52 \pm 1.45$ | $20.25 \pm 0.95$ | - | - | - | - |
| FS-ScanNet Split 1 (1-shot) [90] | $20.34 \pm 1.50$ | $13.64 \pm 0.54$ | - | - | - | - |
| FS-ScanNet Split 1 (3-shot) [90] | $36.75 \pm 1.53$ | $24.42 \pm 0.63$ | - | - | - | - |
| FS-ScanNet Split 1 (5-shot) [90] | $37.45 \pm 1.72$ | $26.54 \pm 0.97$ | - | - | - | - |
| FS-ScanNet Split 2 (1-shot) [90] | $17.23 \pm 1.94$ | $6.37 \pm 1.13$ | - | - | - | - |
| FS-ScanNet Split 2 (3-shot) [90] | $25.63 \pm 1.53$ | $13.54 \pm 0.75$ | - | - | - | - |
| FS-ScanNet Split 2 (5-shot) [90] | $41.36 \pm 1.45$ | $19.75 \pm 0.56$ | - | - | - | - |
| *Grounding 3-D object detection* | | | | | | |
| ScanRefer [5] | - | - | $59.5 \pm 1.7$ | $52.7 \pm 0.9$ | - | - |
| Multi3DRefer [89] | - | - | - | - | $59.4 \pm 1.3$ | $53.8 \pm 0.5$ |
| ViGiL3D [72] | - | - | $15.7 \pm 1.6$ | $13.3 \pm 1.1$ | - | - |
| OpenLex3D (Replica) [28] | - | - | $19.5 \pm 0.9$ | $17.9 \pm 0.6$ | - | - |
| OpenLex3D (ScanNet++) [28] | - | - | - | - | $11.3 \pm 0.7$ | $5.4 \pm 0.4$ |
| OpenLex3D (HM3D) [28] | - | - | $9.9 \pm 0.6$ | $7.6 \pm 0.4$ | - | - |

Table 8: Ablation study on three 3-D detection and grounding benchmarks comparing box initialization strategies. Using average box size as a prior leads to consistent improvements over random initialization.

| Method | ScanNetV2 [65] | | FS-SUNRGBD (1-shot) [90] | | ScanRefer [5] | |
|---|---|---|---|---|---|---|
| | mAP@25 | mAP@50 | mAP@25 | mAP@50 | Acc@25 | Acc@50 |
| Method w/ random initial | $71.2 \pm 1.6$ | $59.5 \pm 0.9$ | $19.52 \pm 1.83$ | $6.49 \pm 0.83$ | $57.2 \pm 1.8$ | $51.4 \pm 1.1$ |
| Method w/ average size of boxes | $72.8 \pm 1.3$ | $60.3 \pm 0.7$ | $20.69 \pm 1.32$ | $6.52 \pm 0.62$ | $59.5 \pm 1.7$ | $52.7 \pm 0.9$ |

ScanNetV2 [65], FS-SUNRGBD [90], and ScanRefer [5]. This confirms that incorporating simple geometric priors leads to more stable and accurate predictions in both detection and grounding tasks.

**Box initialization strategy**    We adopt a strategy that uses the average size of bounding boxes in the dataset to provide a strong prior for initialization. As shown in Table 8, this approach outperforms random initialization across all benchmarks, confirming our hypothesis that informed priors lead to more stable and accurate performance. We also observe that the variance in performance is notably higher under random initialization, which is intuitive since randomly sampled box sizes introduce greater variability in the optimization process.

**Foundation models**    For extracting object query features, we leverage two vision-language foundation models: CLIP2Point [25] and Uni3D [92]. As shown in Table 9, the overall performance is slightly better when using Uni3D across all three benchmarks. While CLIP2Point and Uni3D differ more significantly in zero-shot 3-D scene recognition settings, their impact on object query features is more subtle. This is likely because the downstream detection performance relies more heavily on the diffusion process and latent refinement, rather than the initial feature representation alone.

**Number of object anchors**    Following prior diffusion-based detectors, we adopt 128 initial object anchors as the default setting for a fair comparison. As shown in Table 10, increasing the number of anchors from 64 to 256 leads to improved performance in both general and few-shot 3-D object detection tasks. However, the gains are marginal for grounding tasks such as ScanRefer. This is expected, as ScanRefer typically involves a single target object per query, and 128 anchors already provide sufficient coverage for localizing one bounding box.

Table 9: Ablation study comparing different foundation models for extracting object query features. Using Uni3D leads to slightly better performance than CLIP2Point across three 3-D detection and grounding benchmarks.

| Method | ScanNetV2 [65] | | FS-SUNRGBD (1-shot) [90] | | ScanRefer [5] | |
|---|---|---|---|---|---|---|
| | mAP@25 | mAP@50 | mAP@25 | mAP@50 | Acc@25 | Acc@50 |
| Method w/ CLIP2Point [25] | $72.2 \pm 1.3$ | $59.0 \pm 0.8$ | $19.11 \pm 1.35$ | $6.34 \pm 0.61$ | $58.7 \pm 1.7$ | $51.6 \pm 0.9$ |
| Method w/ Uni3D [92] | $72.8 \pm 1.3$ | $60.3 \pm 0.7$ | $20.69 \pm 1.32$ | $6.52 \pm 0.62$ | $59.5 \pm 1.7$ | $52.7 \pm 0.9$ |

Table 10: Ablation study on the number of object anchors used in the detection pipeline. While increasing the number of anchors improves general and few-shot detection performance, the effect is limited in single-object grounding scenarios such as ScanRefer.

| Method | ScanNetV2 [65] | | FS-SUNRGBD (1-shot) [90] | | ScanRefer [5] | |
|---|---|---|---|---|---|---|
| | mAP@25 | mAP@50 | mAP@25 | mAP@50 | Acc@25 | Acc@50 |
| Method w/ 64 | $69.3 \pm 1.3$ | $56.7 \pm 0.9$ | $17.23 \pm 1.31$ | $5.74 \pm 0.62$ | $57.4 \pm 1.7$ | $51.6 \pm 0.9$ |
| Method w/ 128 | $72.8 \pm 1.3$ | $60.3 \pm 0.7$ | $20.69 \pm 1.32$ | $6.52 \pm 0.62$ | $59.5 \pm 1.7$ | $52.7 \pm 0.9$ |
| Method w/ 256 | $73.2 \pm 1.2$ | $61.2 \pm 0.7$ | $21.45 \pm 1.27$ | $6.67 \pm 0.60$ | $59.7 \pm 1.6$ | $52.8 \pm 0.8$ |

Table 11: Ablation study on loss weighting strategies. The proposed scheduled weights improve stability and generalization across detection tasks, compared to fixed or single-objective settings.

| Loss Weights ($\lambda_{\text{diff}}$, $\lambda_{\text{det}}$) | ScanNetV2 [65] | | FS-SUNRGBD (1-shot) [90] | | ScanRefer [5] | |
|---|---|---|---|---|---|---|
| | mAP@25 | mAP@50 | mAP@25 | mAP@50 | Acc@25 | Acc@50 |
| $(1.0, 0.2) \rightarrow (0.5, 1.0)$ | $72.8 \pm 1.3$ | $60.3 \pm 0.7$ | $20.69 \pm 1.32$ | $6.52 \pm 0.62$ | $59.5 \pm 1.7$ | $52.7 \pm 0.9$ |
| $(1.0, 1.0) \rightarrow (1.0, 1.0)$ | $71.2 \pm 1.4$ | $59.1 \pm 0.8$ | $18.92 \pm 1.51$ | $6.07 \pm 0.60$ | $60.1 \pm 1.5$ | $51.2 \pm 0.8$ |
| $(1.0, 0.2) \rightarrow (1.0, 0.2)$ | $70.3 \pm 1.5$ | $58.4 \pm 0.9$ | $17.53 \pm 1.44$ | $5.71 \pm 0.67$ | $58.4 \pm 1.6$ | $50.7 \pm 0.9$ |
| $(0.0, 1.0) \rightarrow (0.0, 1.0)$ | $66.1 \pm 1.7$ | $53.6 \pm 1.2$ | $12.41 \pm 1.65$ | $4.38 \pm 0.82$ | $54.2 \pm 1.9$ | $47.3 \pm 1.1$ |

**Loss weight scheduling**   We study the impact of different loss weighting strategies between the latent space denoising loss $\mathcal{L}_{\text{diff}}$ and the detection loss $\mathcal{L}_{\text{det}}$. As shown in Table 11, our proposed scheduling scheme starting from $(\lambda_{\text{diff}}, \lambda_{\text{det}}) = (1.0, 0.2)$ and gradually transitioning to $(0.5, 1.0)$ achieves the best overall performance in general, few-shot and grounding benchmarks.

Using equal weights $(1.0, 1.0)$ improves grounding slightly but performs worse in low-data scenarios such as FS-SUNRGBD, suggesting overemphasis on detection loss early in training. Keeping a fixed low weight on detection loss without scheduling, as in $(1.0, 0.2)$, leads to underfitting and degraded overall accuracy. Complete removal of diffusion loss $(0.0, 1.0)$ results in significant performance drops in all tasks, confirming that the latent denoising objective is essential for effective and robust localization.

**Accelerated variants for efficiency**   For completeness, we report two accelerated variants of our method: a 4-step and a 2-step LCM-LoRA [43] model. Due to the limited timeframe for the rebuttal, the newly introduced hyperparameters, such as learning-rate warm-up and cosine decay schedules for the LoRA adapter, have not yet been fully optimized. We anticipate further improvements with more extensive tuning. As shown in Table 12, although our base model is relatively large, its end-to-end runtime matches or exceeds the FPS throughput of competing methods, while consistently achieving higher accuracy. The promptable design also naturally extends to few-shot and grounding tasks without requiring retraining, which highlights the versatility of our framework. In addition, standard engineering optimizations, including mixed precision, quantization, and channel scaling or sparse coding approaches such as Matryoshka representations, could further enhance the efficiency of our method. Nevertheless, we wish to emphasize that the primary focus of this paper is to introduce the promptable concept within a diffusion-based detection framework and to explore the broader potential of diffusion models.

Table 12: Performance of the proposed method and its accelerated variants on general 3D object detection, evaluated on ScanNetV2.

| Method | ScanNetV2 [65] | | | |
|---|---|---|---|---|
| | mAP@50 ($\uparrow$) | Model Parameters ($\downarrow$) | Latency/scene ($\downarrow$) | FPS ($\downarrow$) |
| Diffusion-SS3D [17] | 64.1 | - | - | 30.07 |
| Diff3DETR [10] | 65.7 | - | - | - |
| Ours (DDIM-8) | 72.8 | 1.03B | 0.28s | 3.60 |
| Ours (LCM-LoRA-4 [43]) | 72.3 | 1.03B + 70M LoRA | 0.05s | 22.23 |
| Ours (LCM-LoRA-2 [43]) | 71.9 | 1.03B + 70M LoRA | 0.03s | 30.07 |

## B.3 Qualitative results

Figure 6 presents qualitative examples from the ScanNetV2 and Multi3DRefer datasets. The proposed method produces accurate and well-localized 3-D bounding boxes in both general object detection and grounding tasks, with predicted boxes closely matching the ground truth. These results highlight the model's robustness and adaptability across different types of 3-D scenes.

## C Limitations

While the proposed method demonstrates strong versatility across various 3-D detection tasks, the denoising process inherent to latent diffusion models relies on iterative refinement, which can be computationally expensive. Future work may explore integrating one-step alternatives (*e.g.* Consistency models or Rectified flow models) to improve efficiency. Moreover, although the proposed framework supports promptable conditioning, our experiments primarily focus on text and visual modalities. Leveraging more complex prompts (*e.g.* audio or video) remains an open direction for future research.

## D Broader impacts

The proposed method has the potential to benefit a range of applications in embodied AI, robotics, and interactive world models, especially in scenarios where labeled 3-D data is limited. However, as the approach builds upon foundation models for vision-language alignment, it may inherit biases present in those models. This could lead to skewed predictions when deployed in real world settings.

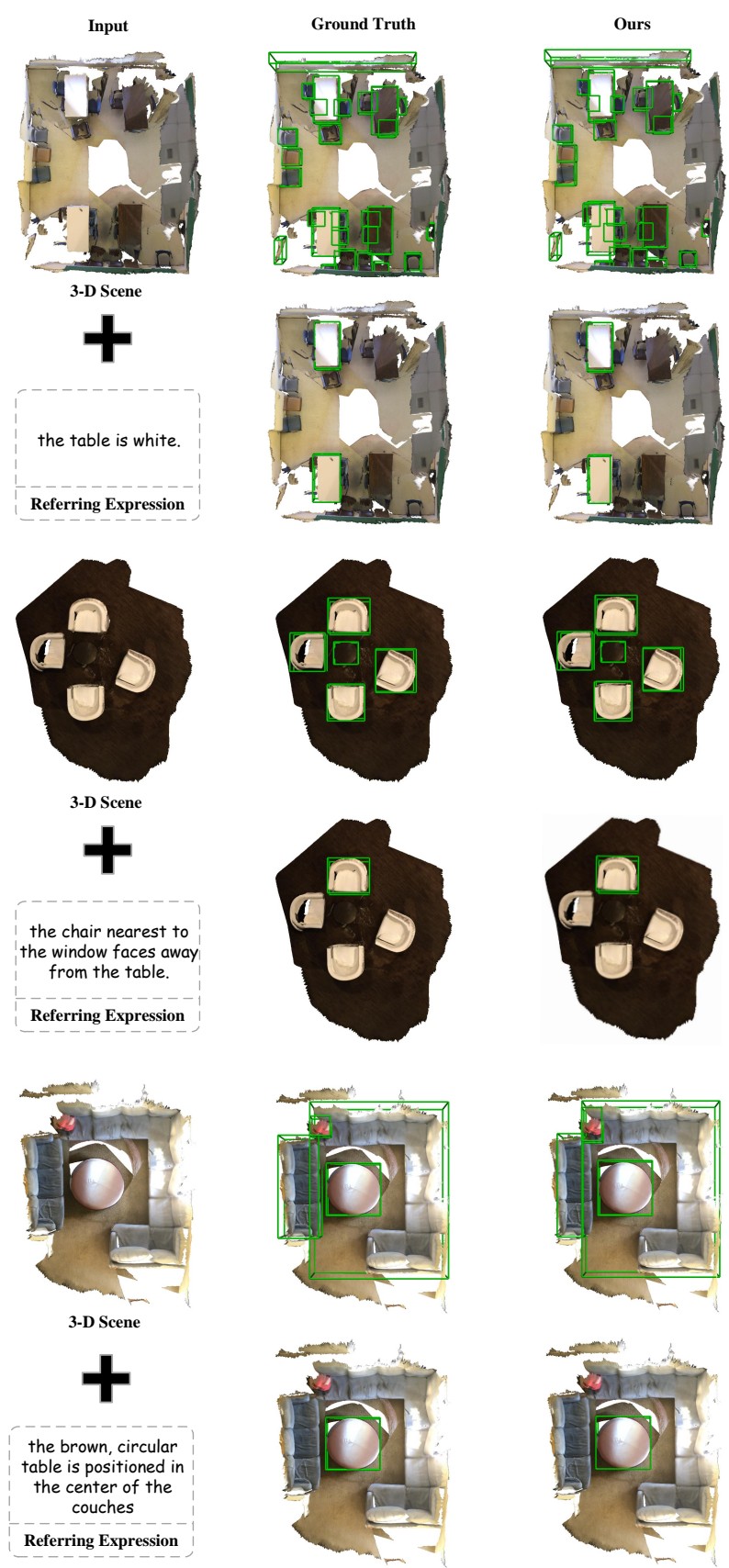

Figure 6: Qualitative results on general 3-D object detection and grounding-based object detection.

