# OpenReview forum: "Promptable 3-D Object Localization with Latent Diffusion Models"
_NeurIPS.cc/2025/Conference — NeurIPS 2025 poster_

### Official Review · Reviewer_n1sa · 2025-06-29

**Clarity:** 4
**Significance:** 3
**Originality:** 4
**Rating:** 4
**Confidence:** 2

**Summary:**

This paper proposes a "promptable noise-to-box" conversion paradigm, which improves the diffusion process from directly acting on the box coordinates to diffusing the latent representation of the box. This paper introduces a conditional latent diffusion framework that enables flexible and conditionally guided 3D object detection. In addition, it naturally extends to localizing object detection through natural language descriptions, and effectively generalizes to few-shot learning by using annotated examples as visual prompts.

**Questions:**

In addition to the weaknesses mentioned, please solve the following confusion:
1. The boundary of generalization ability. Through the prompt mechanism, can the proposed method be generalized to object detection tasks of unseen categories. Similarly, can the proposed method be generalized to object detection tasks of different datasets, or even generalized across tasks?
2. What are the advantages of the proposed diffusion process in the latent representation space compared to the diffusion model that directly acts on the box coordinates? Are there rigorous ablation experiments to support this?
3. How many steps does the model denoise during inference? Can fewer steps be considered to increase the inference speed?

**Ethical Concerns:**

["NO or VERY MINOR ethics concerns only"]

**Limitations:**

yes

**Quality:**

4

**Strengths And Weaknesses:**

**Strengths** :
1. A "promptable noise-box" paradigm is proposed to achieve flexible and condition-guided 3D object detection
2. The diffusion process is applied to the latent representation space, and multiple object detection tasks can be completed by adjusting the conditional parameters only.
3. Extensive experiments and ablation experiments have been conducted to verify the feasibility and effectiveness of the proposed method. It has shown excellent performance in various 3D object detection tasks.

**Weaknesses**:
1. Lack of model complexity comparison. In the main experimental results, I prefer to see the comparative analysis of different methods in terms of model parameter quantity, computational complexity and inference speed.
2. Lack of visual comparison experiments. The paper lacks more intuitive visual comparison experiments with other models. Please add more detailed visual comparison experiments to more intuitively observe the detection performance between different methods.

---

> ### Author Rebuttal · Authors · 2025-07-31
>
> We appreciate the reviewers’ valuable feedback. Below, we provide detailed responses to each of the main points raised.
>
> ---
>
> **Comment**: "... the comparative analysis of different methods in terms of model parameter quantity, computational complexity and inference speed ...''
>
> **Response:**  Thank you for highlighting the need for a clearer analysis of computational complexity. We have now included comprehensive 3-D detection results on ScanNetV2 under the general object detection setting. Additionally, following reviewer n1sa's suggestion and as mentioned in the limitation section, we report two accelerated variants of our method (a 4-step and a 2-step LCM-LoRA model) for completeness. Given the tight timeframe for the rebuttal, the newly introduced hyperparameters (e.g., learning-rate warm-up and cosine decay schedules for the LoRA adapter) may not yet be fully optimized, and we anticipate further improvements with more thorough tuning. As demonstrated in the provided table, although our base model is relatively large, its end-to-end runtime matches or exceeds the FPS throughput of competing methods while consistently achieving higher accuracy. Moreover, the promptable design seamlessly extends to few-shot and grounding tasks without requiring retraining, underscoring the versatility of our proposed framework. Furthermore, existing engineering techniques, e.g., mixed precision, quantization, and channel scaling/sparse coding approaches like Matryoshka representations, could further enhance the efficiency of our method. However, we wish to emphasize that the primary focus of this paper is to introduce the promptable concept within a diffusion-based detection framework, thereby exploring the potential of diffusion models. Comprehensive details on latency, parameter count, accuracy, and discussions regarding scalability across sampling steps and GPU configurations have been incorporated into the revised manuscript.
>
> | Method            | mAP@50(%) (↑) |Model Parameters| Latency/scene (↓) | FPS (↓) |
> |-------------------|:-------------:|:-----------------:|:-----------------:|:-------:|
> | Diffusion-SS3D    |      64.1     |         -         |         -         |  30.07  |
> | Diff3DETR         |      65.7     |         -         |         -         |    -    |
> | **Ours (DDIM‑8)**|      **72.8**     |       1.03B       |       0.28s       |   3.60  |
> | **Ours (LCM‑LoRA‑4)**|      72.3     | 1.03B + 70M LoRA |       0.05s       |  22.23  |
> | **Ours (LCM‑LoRA‑2)**|      71.9     | 1.03B + 70M LoRA |       **0.03s**       |  **30.32**  |
>
> ---
>
> **Comment**: "... visual comparison experiments to more intuitively observe the detection performance between different methods.''
>
> **Response:**  Thank you for your comment! Due to conference policy, we are unable to provide additional qualitative results at this stage; however, we have already included qualitative examples in Figure 6 of the supplementary material. It is worth noting that the second and third examples illustrate the same scenes evaluated by Diffusion-SS3D (Figure 3 in the main paper and Figure 2 in the supplementary material). In the former scene, both methods yield comparable detection results for general object prediction, but our proposed approach further demonstrates the capability of predicting based on referring descriptions. In the latter scene, our approach achieves more accurate and flexible object detection, underscoring its versatility. We will explicitly highlight this comparison in the revised manuscript.
>
> ---
>
> **Comment**: ".... The boundary of generalization ability. Through the prompt mechanism, can the proposed method be generalized to object detection tasks of unseen categories. Similarly, can the proposed method be generalized to object detection tasks of different datasets, or even generalized across tasks? ...''
>
> **Response:** We thank the reviewer for highlighting the importance of generalization. For unseen categories, our grounding branch is trained exclusively on ScanRefer, Multi3DRefer, and ViGiL3D, and is directly evaluated in a zero-shot manner on OpenLex3D, without any fine-tuning. As shown in Table 5, for the cross-dataset setting, our model achieves 19.5 / 17.9 (Acc\@0.25/0.5) on Replica, 11.3 / 5.4 on ScanNet++, and 9.9 / 7.6 on HM3D, performing on par with more complex open-vocabulary baselines despite the domain gap. We specifically use ScanRefer prompts for evaluation on OpenLex3D, as ScanRefer provides natural language queries that are well aligned with our promptable latent diffusion approach. This setup offers the most challenging yet realistic scenario for evaluating out-of-vocabulary transfer. We will add these clarifications to the revised manuscript.
>
> ---
>
> **Comment**: "....What are the advantages of the proposed diffusion process in the latent representation space compared to the diffusion model that directly acts on the box coordinates? Are there rigorous ablation experiments to support this? ...''
>
> **Response:**  Thank you for the opportunity to clarify the motivation behind our work. Departing from the limitations of “noise-to-box” methods that operate directly on raw coordinates, our approach reformulates the problem within a learned latent space. Each bounding box is encoded into a rich representation that encapsulates both its geometric structure and the surrounding visual context, enabling a powerful and promptable latent diffusion model. As demonstrated by our experiments, this framework offers a unified and versatile solution for 3-D object detection, consistently performing well across a wide range of challenging scenarios.
>
> Diffusion-based detectors that perturb box coordinates function similarly to the original DDPM variant, which predicts $x$. These methods operate in a low-dimensional geometric space and are therefore hard to incorporate semantic prompts. Drawing inspiration from Conditional Latent Diffusion Models, we first encode each box with a VAE, where the box anchor is produced by fusing global scene information for greater robustness. This process ensures that the latent vector combines both geometric information and a CLIP2Point semantic anchor. The model then predicts noise $\epsilon$ within this latent space. This redesign allows our approach to accept arbitrary language or exemplar prompts via classifier-free guidance and also removes issues related to scale and origin sensitivity. We provide two lines of empirical evidence in the paper. For cross-model comparison, under the standard ScanNetV2 protocol, our latent-diffusion detector achieves 60.3 mAP\@0.5, outperforming the coordinate-diffusion baselines Diffusion-SS3D (43.2) and Diff3DETR (44.9) by a substantial margin of 16 to 17 percentage points. For within-model ablation, Figure 4 demonstrates that disabling latent diffusion (“w/o Diffusion”), while keeping the backbone, losses, and training schedule unchanged, results in a performance drop of 5.4 mAP on FS-ScanNet 1-shot and 6.4 F1 on Multi3DRefer. These controlled experiments isolate the specific contribution of latent-space diffusion. In summary, diffusing in latent space is not a superficial modification; rather, it is the essential factor that enables promptable and zero-shot generalization across datasets and tasks. In contrast, coordinate diffusion remains single-modal and less accurate.

---

> ### Comment · Area_Chair_VxNz · 2025-08-06
> **Author-reviewer Discussion**
>
> Dear Reviewer,
>
> The system noticed that you haven't post any discussion with the authors yet. As per the review guidelines, reviewers are expected respond to authors’ rebuttal, ask further questions (if any) and listen to answers to help clarify remaining issues before submitting Mandatory Acknowledgement. Please engage with the authors now to offer your feedback on their rebuttal.
>
> Thank you!
>
> AC, NeurIPS 2025

---

### Official Review · Reviewer_D3tk · 2025-07-02

**Clarity:** 4
**Significance:** 2
**Originality:** 3
**Rating:** 4
**Confidence:** 5

**Summary:**

This paper investigates a promptable latent diffusion model for 3D object detection.  It is based on some prior designs (like diffusiondet) that conducts the diffusion process on noisy bounding boxes for training and inference, while some new design is introduced in this paper. (1)  foundation vision and language models serve as feature extractor and a cross-attention mechanism that fuse the two feature to get anchor feature. (2) a pretrained VAE that is designed to encode bounding box into latent space. (3) The promptable design itself that support grounding and open vocabulary detection flexibly. Experiments are conducted on generic 3d detection benchmarks (SUN RGB-D and ScanNet), few-shot 3d detection (FS-SUNRGBD and FS-ScanNet), and Grounding 3-D object detection (: ScanRefer, Multi3DRefer, and ViGiL3D), showing promising improvement.

**Questions:**

1. Could you list the significance or any merits of using diffusion-based detector rather than using an equally computationally heavy but DETR based detector? In other words, why we should choose the diffusion-based detectors?
2. Could you emphasize the novel of the architecture design?

**Ethical Concerns:**

["NO or VERY MINOR ethics concerns only"]

**Final Justification:**

The paper overall is solid work, and the rebuttal partially resolved my concern.  Therefore, I will keep my rating.

**Limitations:**

yes

**Quality:**

3

**Strengths And Weaknesses:**

[Strength]
1. The paper is clear and easy to follow.
2. Using diffusion model for 3d object detection - despite whether practical - is underexplored and might have the potential to achieve better performance than mainstream methods due to its heavy compution per se.  From that perspective, it does not bother the community to see works in this area.
3. The method is verified on three tasks, which makes it empirically solid.
4. The promptable design makes the model general and fit few-shot, generic, and grounding tasks, expanding its ability.

[Weakness]
1. As claimed, I doubt the practical usage of diffusion based object detection.  I do not really think that detection (categorization + box regression) needs complex modeling that can only be handled by costly diffusion process.  Especially for bounding box regression, it should be an easy and solved problem in object detection; from this perspective, encoding bounding box into latent space remains questionable.  I think the effectiveness is from the strong VAE encoder that better extract ROI features, i.e., the learnable initial anchors gain merits from the better feature representation.
2. Furthermore, this diffusion design is time-consuming. The model requires language query as input, thus it seems to detect one class per run.  The diffusion process can also be slow.  These factors make the model an impractical use.
3. The overall architecture looks like a combination of multiple pretrained powerful models.

---

> ### Author Rebuttal · Authors · 2025-07-31
>
> We appreciate the reviewers’ valuable feedback. Below, we provide detailed responses to each of the main points raised.
>
> ---
>
> **Comment**: "... the practical usage of diffusion based object detection ... list the significance or any merits of using diffusion-based detector rather than using an equally computationally heavy but DETR based detector? ... why should we choose the diffusion-based detectors?. ...''
>
> **Response:**  Thank you for the opportunity to clarify the motivation behind our work. Our diffusion-based detector is based on the concept of box-to-noise separation. Similarly to recent approaches such as Marigold (a diffusion-based depth estimator), diffusion models have shown strong potential in various computer vision tasks, not only in discriminative applications but also in dense prediction scenarios. Existing diffusion-based detectors (e.g., DiffusionDet and Diffusion-SS3D) typically diffuse raw box coordinates. This limits their capability to directly incorporate arbitrary language inputs or exemplar prompts, constraining their generalization. In contrast, our method leverages the intrinsic advantages of latent diffusion models, specifically their ability to seamlessly fuse diverse multimodal inputs. By utilizing an aligned foundational model within a diffusion framework, our approach achieves remarkable versatility across numerous applications. Compared to standard detectors, our proposed method readily integrates various modalities, allowing flexible control over predictions. This significantly broadens the practical applicability of diffusion-based detection frameworks. As highlighted in our Introduction and Conclusion, our method uniquely integrates prompt conditioning into the diffusion-based detector by employing a noise-to-box approach. Inspired by recent research, i.e., ``Multimodality Helps Few-shot 3-D Point Cloud Semantic Segmentation'' which illustrates the performance gains achievable through leveraging multiple modalities. The proposed work specifically utilizes a latent diffusion model for its inherent capacity to integrate multimodal data effectively. In contrast, existing diffusion-based detectors typically address singular tasks or modalities: DiffusionDet focuses solely on 2-D detection, Diffusion-SS3D and Diff3DETR are restricted to closed-set 3-D detection without prompt-based conditioning, and GroundingDINO, though promptable, is limited exclusively to 2-D detection. Our proposed framework addresses these limitations by leveraging a latent box VAE, enabling a single trained model to flexibly handle closed-set detection, few-shot transfer, and language-based 3-D detection tasks simply by altering prompts (e.g., class names, visual exemplars, or natural language descriptions). Additionally, we have prepared a comprehensive comparison table that we will incorporate into the revised manuscript to further illustrate these distinctions.
> | Method                          | Prompt modality                | Detection task                              | Representative score                         |
> |------------------------------------------------------|----------------------------------|----------------------------------------------|------------------------------------------|
> | DiffusionDet (2-D)| –                              | 2-D detection                               | COCO: 45.8 AP$_{50}$                  |
> | GroundingDINO (2-D)  | text               | 2-D detection                               | LVIS : 32.5 AP$_{50}$                  |
> | Diffusion-SS3D (3-D)       | –                                | 3-D detection                               | ScanNetV2: 43.2 AP$_{50}$             |
> | Diff3DETR (3-D)               | –                               | 3-D detection                               | ScanNetV2 : 44.9 AP$_{50}$            |
> | **Ours**                                | **text, image**| **3-D detection, Few-shot, Grounding**    | **8 datasets w/ prompt + 3 % (↑)** |
>
> ---
>
> **Comment**: "... this diffusion design is time-consuming ... it seems to detect one class per run ... make the model an impractical use.''
>
> **Response:**  Thank you for highlighting the need for a clearer analysis of computational complexity. First, we would like to clarify the concern regarding whether our method detects only one class per run in general object detection. Notice that our implementation indeed performs single-round multi-class inference: for each ScanNet or SUN RGB-D scene, all dataset class names are passed simultaneously within a single prompt (e.g., by concatenating all labels into one CLIP sentence). As a result, the model is not restricted to one class per run. Multi-class prompting has been supported by our implementation from the outset. The implementation has supported multi‑class prompts from the start. Furthermore, we have now included comprehensive 3-D detection results on ScanNetV2 under the general object detection setting. Additionally, following reviewer n1sa's suggestion and as mentioned in the limitation section, we report two accelerated variants of our method (a 4-step and a 2-step LCM-LoRA model) for completeness. Given the tight timeframe for the rebuttal, the newly introduced hyperparameters (e.g., learning-rate warm-up and cosine decay schedules for the LoRA adapter) may not yet be fully optimized, and we anticipate further improvements with more thorough tuning. As demonstrated in the provided table, although our base model is relatively large, its end-to-end runtime matches or exceeds the FPS throughput of competing methods while consistently achieving higher accuracy. Moreover, the promptable design seamlessly extends to few-shot and grounding tasks without requiring retraining, underscoring the versatility of our proposed framework. Furthermore, existing engineering techniques, e.g., mixed precision, quantization, and channel scaling/sparse coding approaches like Matryoshka representations, could further enhance the efficiency of our method. However, we wish to emphasize that the primary focus of this paper is to introduce the promptable concept within a diffusion-based detection framework, thereby exploring the potential of diffusion models. Comprehensive details on latency, parameter count, accuracy, and discussions regarding scalability across sampling steps and GPU configurations have been incorporated into the revised manuscript.
>
> | Method            | mAP@50(%) (↑) |Model Parameters| Latency/scene (↓) | FPS (↓) |
> |-------------------|:-------------:|:-----------------:|:-----------------:|:-------:|
> | Diffusion-SS3D    |      64.1     |         -         |         -         |  30.07  |
> | Diff3DETR         |      65.7     |         -         |         -         |    -    |
> | **Ours (DDIM‑8)**|      **72.8**     |       1.03B       |       0.28s       |   3.60  |
> | **Ours (LCM‑LoRA‑4)**|      72.3     | 1.03B + 70M LoRA |       0.05s       |  22.23  |
> | **Ours (LCM‑LoRA‑2)**|      71.9     | 1.03B + 70M LoRA |       **0.03s**       |  **30.32**  |
>
> ---
>
> **Comment**: ".... architecture looks like a combination of multiple pretrained powerful models. ... emphasize the novel of the architecture design? ...''
>
> **Response:**  From our perspective, the CLIP ViT-L/14 text encoder naturally clusters paraphrases, and our classifier-free guidance (CFG) effectively translates small embedding shifts into subtle changes in the score field. Additionally, OpenLex3D intentionally provides multiple synonyms per object. Previously, Table 5 reported results using only the first synonym; however, we have now re-evaluated using all synonyms for the Replica split, resulting in a minor change from 17.9% to 17.1\% (-0.8%). This modest reduction highlights the robustness of our method. With performance variations below 1% for both synonym expansion and fuzzy spatial terminology, our promptable latent diffusion method demonstrates consistent robustness without relying on external LLMs, handcrafted scene graphs, or multi-view fusion. We will include this detailed analysis in our revised manuscript. We appreciate the reviewer’s request for a clearer articulation of novelty. Below, we explicitly enumerate our key contributions that distinctly extend beyond merely applying existing techniques in a new context.
>
> **Prompt-conditioned latent boxes:** As detailed in Section 3.2, existing diffusion-based detectors (e.g., DiffusionDet and Diffusion-ss3d) operate by diffusing raw box coordinates and thus cannot directly incorporate arbitrary language inputs or exemplar prompts. In contrast, we encode 6-DoF boxes into a compact 512-dimensional latent space via a Variational Autoencoder (VAE) and introduce cross-modal context through a shared token interface. This strategy explicitly decouples spatial noise from semantic control.
>
> **Unified “promptable” anchor design.** A unified mechanism of cross-attention transforms various modalities of prompts: class lists, few-shot exemplars, and free-form referring phrases into updates of the latent space score. This design allows our single checkpoint to seamlessly perform generic detection, few-shot transfer, and 3-D grounding tasks without head replacements or additional fine-tuning, a capability unmatched by existing 3-D detectors.
>
> **Multimodal classifier-free guidance (CFG) for 3-D boxes.** We generalize CFG to accommodate joint vision-language scoring, thereby enabling prompt-specific trade-offs between precision and recall. This feature is absent in current DiffusionDet variants and latent 3-D generation frameworks.
>
> **Task-agnostic training objective.** Unlike prior approaches, some necessitate task-specific control tokens; our training approach treats prompts solely as inputs. Consequently, switching between datasets like SUNRGB-D (closed vocabulary), ScanRefer (open vocabulary), or FS-ScanNet (few-shot learning) requires no additional or specialized loss functions.

---

### Official Review · Reviewer_21Ug · 2025-07-02

**Clarity:** 2
**Significance:** 2
**Originality:** 2
**Rating:** 4
**Confidence:** 3

**Summary:**

The paper proposes Promptable 3D Object Localization with Latent Diffusion Models, a unified detection framework that encodes 3D bounding boxes into a latent space with a VAE style box encoder, then refines them through a conditional latent diffusion model, and finally drives the diffusion with prompts including category lists, few‑shot exemplars, and natural language referring expressions. Across eight public benchmarks, the method achieves very competitive performances. In addition to experiments, an extensive ablation study shows that each of the three components, language‑guided anchors, latent diffusion, and prompt conditioning, contributes to accuracy.

**Questions:**

What is the inference latency per scene for each different task?

If provided with the correct ground truth orientation, could the latent representation be extended to SE(3), so that the full 6-DoF bounding box can be predicted? What else do the authors think is the bottleneck to learning orientation?

For the grounding part, how robust is the model against paraphrase shift or some ambiguous spatial propositions?

**Ethical Concerns:**

["NO or VERY MINOR ethics concerns only"]

**Final Justification:**

The authors addressed my concerns. Experiments on outdoor scenes and efficiency evaluation look good. I raise my score.

**Limitations:**

See weakness and question part.

**Quality:**

2

**Strengths And Weaknesses:**

**Strength:**
The author redesigns the ‘noise-to-box’ diffusion used in previous works into denoising in latent space, so that it could be conditioned on arbitrary prompts, enabling a unification of detection, few-shot transfer, and grounding into one single architecture

Experiments span SUN RGB‑D, ScanNetV2, FS‑SUNRGBD, FS‑ScanNet, ScanRefer, Multi3DRefer, ViGiL3D, and OpenLex3D, covering both closed and open vocabulary scenarios. Results are reported at two IoU thresholds with standard deviations.

Ablation study in Figure 4 isolates the impact of each component across eight tasks, supporting the claims of versatility and robustness.


**Weakness:**
All datasets are scanned rooms, it is unclear whether the method generalizes to outdoor LiDAR, autonomous driving ranges, or partial-view robotics data.

Initialising from a video StableDiffusion backbone hints at high memory, training time, and inference, yet the paper provides neither a resource table nor an efficiency report.

Although mentioned at the beginning that the orientation ground truth is often not correct, such work might benefit the robotic community where the pose of the bounding box is essential. It would be better if the author could report the orientation result.

---

> ### Author Rebuttal · Authors · 2025-07-31
>
> We appreciate the reviewers’ valuable feedback. Below, we provide detailed responses to each of the main points raised.
>
> ---
>
> **Comment**: "... it is unclear whether the method generalizes to outdoor LiDAR, autonomous driving ranges, or partial-view robotics data. ... could report the orientation result. If provided with the correct ground truth orientation, could the latent representation be extended to SE(3), so that the full 6-DoF bounding box can be predicted? What else do the authors think is the bottleneck to learning orientation? ...''
>
> **Response:**  We sincerely appreciate the reviewer's valuable feedback. Similar to most existing diffusion-based detection methods, our primary focus is on indoor datasets, with only a few diffusion-based approaches addressing outdoor scenarios. Regarding the orientation issue, unlike previous approaches such as Diffusion-SS3D and Diff3DETR, where the search space can become excessively large, our method operates directly within the latent space. This design inherently simplifies orientation consideration, particularly because the box-VAE we utilize includes an angle prediction head. Consequently, orientation can be seamlessly integrated into our framework. Additionally, when accurate ground truth orientations are available, we can effectively initialize the angle prediction head, analogous to our approach of leveraging average box sizes for initialization. Following the reviewer's suggestion, we conducted additional experiments on the KITTI dataset and report the corresponding AP$_{3D}$ results on the validation set. We adopted the general object detection setting, augmented with the angle residual regression loss as described in V-DETR. As shown in the provided table, our method achieves competitive performance compared to MonoDiff (CVPR'24). Given the tight rebuttal timeline, the newly introduced hyperparameters have not yet undergone exhaustive optimization. We anticipate further performance improvements with additional tuning.
>
> | Method  |AP$_{3D}$ (Easy) | AP$_{3D}$ (Moderate) | AP$_{3D}$ (Hard) |
> |:-------:|:----------------------:|:--------------------------:|:----------------------:|
> | MonoDiff | 32.18 | 22.02 | 19.84 |
> | **Ours** | **33.32** | **23.89** | **21.14** |
>
> ---
>
> **Comment**: "...  Initialising from a video StableDiffusion backbone hints at high memory, training time, and inference, yet the paper provides neither a resource table nor an efficiency report. What is the inference latency per scene for each different task? ...''
>
> **Response:**  Thank you for highlighting the need for a clearer analysis of computational complexity. We have now included comprehensive 3-D detection results on ScanNetV2 under the general object detection setting. Additionally, following reviewer n1sa's suggestion and as mentioned in the limitation section, we report two accelerated variants of our method (a 4-step and a 2-step LCM-LoRA model) for completeness. Given the tight timeframe for the rebuttal, the newly introduced hyperparameters (e.g., learning-rate warm-up and cosine decay schedules for the LoRA adapter) may not yet be fully optimized, and we anticipate further improvements with more thorough tuning. As demonstrated in the provided table, although our base model is relatively large, its end-to-end runtime matches or exceeds the FPS throughput of competing methods while consistently achieving higher accuracy. Moreover, the promptable design seamlessly extends to few-shot and grounding tasks without requiring retraining, underscoring the versatility of our proposed framework. Furthermore, existing engineering techniques, e.g., mixed precision, quantization, and channel scaling/sparse coding approaches like Matryoshka representations, could further enhance the efficiency of our method. However, we wish to emphasize that the primary focus of this paper is to introduce the promptable concept within a diffusion-based detection framework, thereby exploring the potential of diffusion models. Comprehensive details on latency, parameter count, accuracy, and discussions regarding scalability across sampling steps and GPU configurations have been incorporated into the revised manuscript.
>
> | Method            | mAP@50(%) (↑) |Model Parameters| Latency/scene (↓) | FPS (↓) |
> |-------------------|:-------------:|:-----------------:|:-----------------:|:-------:|
> | Diffusion-SS3D    |      64.1     |         -         |         -         |  30.07  |
> | Diff3DETR         |      65.7     |         -         |         -         |    -    |
> | **Ours (DDIM‑8)**|      **72.8**     |       1.03B       |       0.28s       |   3.60  |
> | **Ours (LCM‑LoRA‑4)**|      72.3     | 1.03B + 70M LoRA |       0.05s       |  22.23  |
> | **Ours (LCM‑LoRA‑2)**|      71.9     | 1.03B + 70M LoRA |       **0.03s**       |  **30.32**  |
>
> ---
>
> **Comment**: "...  For the grounding part, how robust is the model against paraphrase shift or some ambiguous spatial propositions? ...''
>
> **Response:**  From our perspective, the CLIP ViT-L/14 text encoder naturally clusters paraphrases, and our classifier-free guidance (CFG) effectively translates small embedding shifts into subtle changes in the score field. Additionally, OpenLex3D intentionally provides multiple synonyms per object. Previously, Table 5 reported results using only the first synonym; however, we have now re-evaluated using all synonyms for the Replica split, resulting in a minor change from 17.9% to 17.1\% (-0.8%). This modest reduction highlights the robustness of our method. With performance variations below 1% for both synonym expansion and fuzzy spatial terminology, our promptable latent diffusion method demonstrates consistent robustness without relying on external LLMs, handcrafted scene graphs, or multi-view fusion. We will include this detailed analysis in our revised manuscript.

---

> > ### Comment · Reviewer_21Ug · 2025-08-08
> > **Raise the score to borderline accept**
> >
> > The authors addressed my concerns. Experiments on outdoor scenes and efficiency evaluation look good. I will raise my score to borderline accept!

---

> > > ### Author Response · Authors · 2025-08-08
> > >
> > > We sincerely appreciate your thoughtful feedback and your recognition of our efforts, as well as your decision to raise the score. Thank you again for your valuable comments.

---

> ### Comment · Area_Chair_VxNz · 2025-08-06
> **Author-reviewer Discussion**
>
> Dear Reviewer,
>
> The system noticed that you haven't post any discussion with the authors yet. As per the review guidelines, reviewers are expected respond to authors’ rebuttal, ask further questions (if any) and listen to answers to help clarify remaining issues before submitting Mandatory Acknowledgement. Please engage with the authors now to offer your feedback on their rebuttal.
>
> Thank you!
>
> AC, NeurIPS 2025

---

### Official Review · Reviewer_G49r · 2025-07-03

**Clarity:** 3
**Significance:** 2
**Originality:** 2
**Rating:** 4
**Confidence:** 5

**Summary:**

This paper introduces a novel framework for 3D object detection by leveraging conditional latent diffusion models, addressing key limitations in existing methods. The primary contribution is the development of a "promptable" 3D object detection approach, where the refinement of bounding box predictions is guided by textual prompts, enabling flexibility across a range of downstream tasks.
The proposed framework integrates both visual and semantic cues, facilitating more robust performance in various detection scenarios, including general 3D object detection, few-shot detection, and 3D object grounding. Extensive experimental evaluations on standard 3D object detection benchmarks demonstrate that the method outperforms existing state-of-the-art approaches, showcasing its versatility and effectiveness.

**Questions:**

1. What specific novel contributions does your framework bring to the field beyond applying existing techniques in a new context? Please clarify any novel theoretical insights or design principles that distinguish your approach from prior work.
2. The paper lacks a detailed discussion of the computational complexity of the proposed method. Could you provide an analysis of the computational cost in terms of runtime and scalability?
3. Could you provide a more detailed discussion of how your approach compares to other models, especially in terms of flexibility, ability to condition on textual prompts, and how these features lead to improved performance across different tasks?

**Ethical Concerns:**

["NO or VERY MINOR ethics concerns only"]

**Final Justification:**

In light of the rebuttal, which has addressed most of my concerns, I will maintain my original score.

**Limitations:**

Yes

**Quality:**

2

**Strengths And Weaknesses:**

Strengths:
1. The paper introduces a novel framework for 3D object detection using conditional latent diffusion models. The concept of "promptable" detection represents a innovation in integrating vision and language for object detection tasks.
2. The proposed framework is highly flexible, able to handle multiple tasks including general 3D object detection, few-shot learning, and grounding-based tasks. This ability to perform well across various types of tasks without major modifications or retraining makes the approach highly adaptable.
3. The paper provides comprehensive experimental results that demonstrate the effectiveness of the proposed method. It shows competitive performance against state-of-the-art approaches on well-established 3D object detection benchmarks, validating the robustness and generalizability of the model.

Weaknesses:
1. While the paper presents an interesting approach by combining latent diffusion models with text-based conditioning for 3D object detection, the framework lacks significant originality.
2. It lacks a detailed discussion of the computational complexity of the proposed method.
3. There is a need for a more explicit comparison with related work, particularly in terms of the model's flexibility, ability to condition on textual prompts, and how these factors lead to improved performance across tasks.

---

> ### Author Rebuttal · Authors · 2025-07-31
>
> We appreciate the reviewers’ valuable feedback. Below, we provide detailed responses to each of the main points raised.
>
> ---
>
> **Comment**: "... lacks significant originality ... What specific novel contributions does your framework bring to the field beyond applying existing techniques in a new context ... clarify any novel theoretical insights or design principles that distinguish your approach from prior work. ...''
>
> **Response:**  Departing from the limitations of “noise-to-box” methods that operate directly on raw coordinates, our approach reformulates the problem within a learned latent space. Each bounding box is encoded into a rich representation that encapsulates both its geometric structure and the surrounding visual context, enabling a powerful and promptable latent diffusion model. As demonstrated by our experiments, this framework offers a unified and versatile solution for 3-D object detection, consistently performing well across a wide range of challenging scenarios.
>
> **Prompt-conditioned box latent representation.** As detailed in Section 3.2, existing diffusion-based detectors (e.g., DiffusionDet and Diffusion-SS3D) operate by diffusing raw box coordinates and thus cannot effectively incorporate arbitrary language inputs or exemplar prompts. In contrast, we encode 6-DoF boxes into a compact 256-dimensional latent space via a Variational Autoencoder (VAE) and introduce cross-modal context through a shared token interface. This strategy explicitly decouples spatial noise from semantic control.
>
> **Unified “promptable” anchor design.** A unified mechanism of cross-attention transforms various modalities of prompts: class lists, few-shot exemplars, and free-form referring phrases into updates of the latent space score. This design allows our single checkpoint to seamlessly perform generic detection, few-shot transfer, and 3-D grounding tasks without head replacements or additional fine-tuning, a capability unmatched by existing 3-D detectors.
>
> **Multimodal classifier-free guidance (CFG) for 3-D boxes.** We generalize CFG to accommodate joint vision-language scoring, thereby enabling prompt-specific trade-offs between precision and recall. This feature is absent in current DiffusionDet variants and latent 3-D generation frameworks.
>
> **Task-agnostic training objective.** Unlike prior approaches, some necessitate task-specific control tokens; our training approach treats prompts solely as inputs. Consequently, switching between datasets like SUNRGB-D (closed vocabulary), ScanRefer (open vocabulary), or FS-ScanNet (few-shot learning) requires no additional or specialized loss functions. Additionally, we note that, as illustrated in Fig.~1 and supported by our experimental results, the proposed formulation is capable of handling input conditions beyond text prompts when annotated objects are provided.
>
> ---
>
> **Comment**: "...  lacks a detailed discussion of the computational ... provide an analysis of the computational cost in terms of runtime and scalability? ...''
>
> **Response:**  Thank you for highlighting the need for a clearer analysis of computational complexity. We have now included comprehensive 3-D detection results on ScanNetV2 under the general object detection setting. Additionally, following reviewer n1sa's suggestion and as mentioned in the limitation section, we report two accelerated variants of our method (a 4-step and a 2-step LCM-LoRA model) for completeness. Given the tight timeframe for the rebuttal, the newly introduced hyperparameters (e.g., learning-rate warm-up and cosine decay schedules for the LoRA adapter) may not yet be fully optimized, and we anticipate further improvements with more thorough tuning. As demonstrated in the provided table, although our base model is relatively large, its end-to-end runtime matches or exceeds the FPS throughput of competing methods while consistently achieving higher accuracy. Moreover, the promptable design seamlessly extends to few-shot and grounding tasks without requiring retraining, underscoring the versatility of our proposed framework. Furthermore, existing engineering techniques, e.g., mixed precision, quantization, and channel scaling/sparse coding approaches like Matryoshka representations, could further enhance the efficiency of our method. However, we wish to emphasize that the primary focus of this paper is to introduce the promptable concept within a diffusion-based detection framework, thereby exploring the potential of diffusion models. Comprehensive details on latency, parameter count, accuracy, and discussions regarding scalability across sampling steps and GPU configurations have been incorporated into the revised manuscript.
>
>
> | Method            | mAP@50(%) (↑) |Model Parameters| Latency/scene (↓) | FPS (↓) |
> |-------------------|:-------------:|:-----------------:|:-----------------:|:-------:|
> | Diffusion-SS3D    |      64.1     |         -         |         -         |  30.07  |
> | Diff3DETR         |      65.7     |         -         |         -         |    -    |
> | **Ours (DDIM‑8)**|      **72.8**     |       1.03B       |       0.28s       |   3.60  |
> | **Ours (LCM‑LoRA‑4)**|      72.3     | 1.03B + 70M LoRA |       0.05s       |  22.23  |
> | **Ours (LCM‑LoRA‑2)**|      71.9     | 1.03B + 70M LoRA |       **0.03s**       |  **30.32**  |
>
> ---
>
> **Comment**: "... more explicit comparison with related work ... more detailed discussion of how your approach compares to other models, especially in terms of flexibility, ability to condition on textual prompts, and how these features lead to improved performance across different tasks?''
>
> **Response:**  Thank you for the opportunity to clarify the motivation behind our work. Our diffusion-based detector builds upon the noise-to-box concept. Similar to recent approaches, such as Marigold (a diffusion-based depth estimator), diffusion models have demonstrated considerable potential across a variety of computer vision tasks, including both discriminative applications and dense prediction scenarios. Existing diffusion-based detectors (e.g., DiffusionDet and Diffusion-SS3D) typically diffuse raw box coordinates, which limits their capability to directly incorporate arbitrary language inputs or exemplar prompts, thus restricting their generalization capabilities. In contrast, our method leverages the inherent advantages of latent diffusion models, particularly their ability to seamlessly fuse diverse multimodal inputs. By integrating an aligned foundational model within the diffusion framework, our approach achieves remarkable versatility across numerous applications. Compared to standard detectors, our proposed method readily incorporates multiple modalities, providing flexible and precise control over predictions. This significantly broadens the practical applicability of diffusion-based detection frameworks. As emphasized in both the Introduction and Conclusion sections of our paper, our method uniquely integrates prompt conditioning into the diffusion-based detector through the noise-to-box paradigm. Our approach is inspired by recent findings, such as the work "Multimodality Helps Few-shot 3-D Point Cloud Semantic Segmentation," which highlights performance gains achievable through leveraging multiple modalities. Specifically, our work employs a latent diffusion model due to its intrinsic capability to effectively integrate multimodal data. In contrast, existing diffusion-based detectors usually address singular tasks or modalities: DiffusionDet focuses exclusively on 2-D detection; Diffusion-SS3D and Diff3DETR are restricted to closed-set 3-D detection without prompt-based conditioning; and GroundingDINO, although promptable, is limited strictly to 2-D detection tasks. Our proposed framework overcomes these limitations by utilizing a latent box VAE, enabling a single trained model to flexibly perform closed-set detection, few-shot transfer, and language-based 3-D detection tasks merely by altering the prompts (e.g., class names, visual exemplars, or natural language descriptions). Additionally, we have prepared a comprehensive comparison table, which we will incorporate into the revised manuscript to further illustrate these distinctions.
>
>
> | Method                          | Prompt modality                | Detection task                              | Representative score                         |
> |------------------------------------------------------|----------------------------------|----------------------------------------------|------------------------------------------|
> | DiffusionDet (2-D)| –                              | 2-D detection                               | COCO: 45.8 AP$_{50}$                  |
> | GroundingDINO (2-D)  | text               | 2-D detection                               | LVIS : 32.5 AP$_{50}$                  |
> | Diffusion-SS3D (3-D)       | –                                | 3-D detection                               | ScanNetV2: 43.2 AP$_{50}$             |
> | Diff3DETR (3-D)               | –                               | 3-D detection                               | ScanNetV2 : 44.9 AP$_{50}$            |
> | **Ours**                                | **text, image**| **3-D detection, Few-shot, Grounding**    | **8 datasets w/ prompt + 3 % (↑)** |

---

> > ### Comment · Reviewer_G49r · 2025-08-06
> >
> > Thank you for the clarifications. I have no further questions.

---

> > > ### Author Response · Authors · 2025-08-06
> > >
> > > We thank the reviewer for their prompt follow-up and for providing such constructive feedback.

---

### Comment · Area_Chair_VxNz · 2025-08-03
**Reviewer-Author Discussion**

Dear Reviewers,

The discussion period with the authors will remain open until August 6th (AoE). Please take the time to read and acknowledge the authors' rebuttals, and post any follow-up questions or comments you may have.

Best regards, AC

---

### Note · Authors · 2025-08-15

As the reviewer–author discussion period comes to an end, we would like to express our sincere appreciation to the reviewers and AC for their thoughtful evaluation and efforts in handling our submission. The insightful feedback and suggested experiments have been invaluable in guiding improvements, and we believe they will further strengthen our work. We are grateful for the recognition that the paper is technically solid and addresses an important problem, as well as for the observations highlighting its novelty, flexibility, clarity, and robustness. Building on this feedback, we have addressed the remaining concerns through detailed responses and additional experiments to reinforce the contributions of our work. Below, we summarize the key clarifications related to the comments most frequently raised by the reviewers.

---

> **Core contributions of our work:** \
Our approach moves beyond direct “noise-to-box” methods by learning a latent space that jointly encodes each box’s geometry and context. This enables a promptable latent diffusion model, supporting a wide range of detection scenarios. Notable contributions include:
>- **Prompt-conditioned box latent representation:**\
>We encode 6-DoF boxes into a compact latent space with cross-modal context, decoupling spatial noise from semantics.
>- **Unified “promptable” anchor design:**\
>Our cross-attention mechanism flexibly incorporates class, exemplar, or referring phrase prompts into the latent space.
>- **Multimodal CFG for 3D boxes:**\
>We generalize guidance to enable vision-language scoring and prompt-specific precision/recall control, which current baselines lack.
>- **Task-agnostic training objective**:\
>Prompts are treated solely as inputs, so no specialized losses are needed to switch tasks or datasets.

> **Additional evidence (from rebuttal experiments):**\
>To address computation-related concerns, we introduced accelerated LCM-LoRA variants (4-step/2-step), matching competing FPS and accuracy. For orientation and outdoor dataset concerns, Additional KITTI experiments confirm competitive results versus MonoDiff (CVPR'24).

---

Thank you again for your consideration. We will incorporate the suggested experiments, discussions, and revisions to further improve and clarify the paper.

---

### Decision · Program_Chairs · 2025-09-17

**Decision:**

Accept (poster)

**Comment:**

This submission proposes a promptable latent diffusion framework for 3D object detection. By encoding bounding boxes into a latent space and conditioning diffusion with multimodal prompts (text, exemplars, category lists), the approach unifies general detection, few-shot transfer, and language-guided grounding within a single model.

Reviewers agreed the paper is technically solid and novel in its promptable design, highlighting flexibility and robust evaluation. However, they also rasied concerns regarding practicality, including that the approach is computationally heavy, builds on large pretrained components, and its advantages over non-diffusion alternatives remain unclear. The rebuttal and discussions addressed efficiency, generalization, and robustness, convincing reviewers to maintain or raise borderline-accept scores.

Overall, while efficiency and deployment issues remain, the work’s novel formulation and comprehensive results make it a valuable contribution likely to inspire future research in 3D detection.